# OBJECT-LEVEL SELF-DISTILLATION WITH BOUNDING BOXES IMPROVES VISION PRETRAINING

## ABSTRACT

Self-distillation has become a central paradigm for pretraining vision transformers without labels. Existing approaches typically operate at the image level and assume that different augmentations of an image preserve the same semantic content. This assumption often fails in complex scenes with multiple objects, where random crops possibly contain different semantic content. To address this, we propose to enrich the learning signal by incorporating object-level annotations. Using object bounding boxes as weak-supervision, we introduce ODIS (Object-level Self-Distillation), a new framework that refines the self-distillation objective to the level of individual objects. ODIS uses object-aware cropping to ensure that teacher and student views depict the same object, and employs masked attention to focus the learning signal on that object. On ImageNet-1K, ODIS learns stronger visual representations than image-level distillation methods such as iBOT across both image- and patch-level benchmarks, and its features transfer better to downstream classification and retrieval tasks. Our results highlight the importance of object-centric supervision in scalable representation learning and show how object-level annotations can be integrated into distillation pipelines to enhance generalization.

## 1 INTRODUCTION

Vision Transformers (ViTs) (Dosovitskiy et al., 2020) trained with self-distillation (Caron et al., 2021; Oquab et al., 2023) achieve state-of-the-art results across many tasks, from unsupervised segmentation to dense correspondence and appearance transfer (Amir et al., 2021; Tumanyan et al., 2022; Ofri-Amar et al., 2023; Hamilton et al., 2022). As with large language models, performance improves with scaling model size, compute, and pretraining data (Oquab et al., 2023; Siméoni et al., 2025).

However, scaling data alone does not guarantee better representations or downstream performance (Goyal et al., 2021; Oquab et al., 2023; Siméoni et al., 2024; Vo et al., 2024). For example, DINOv3 shows that pretraining on 17B raw images yields 4.5% lower ImageNet-1K $k$-NN accuracy than training the same model on 1.7B curated images (Siméoni et al., 2025, Table 1). As a result, large-scale visual pretraining often incorporates data curation steps to improve pretraining data, such as semi-automatic labeling (Dehghani et al., 2023) or filtering low-quality images using embeddings from pretrained models (Oquab et al., 2023; Siméoni et al., 2025). Inspired by the previous work, we propose to strengthen the pretraining learning signal, but rather than relying only on image-level annotations or filtering, we incorporate *object-level annotations*.

To see why object-level annotations improve learning signal, we first recall the self-distillation objective. In self-distillation, pretraining adopts a teacher-student architecture: the student is trained to match the mean-teacher's outputs rather than dataset labels Tarvainen & Valpola (2017); Grill et al. (2020); Caron et al. (2021) (Fig. 1). At the image level, this is implemented as single-label classification on a global `[CLS]` embedding. This setup assumes the teacher and student receive inputs with the same *content*. This assumption, how-

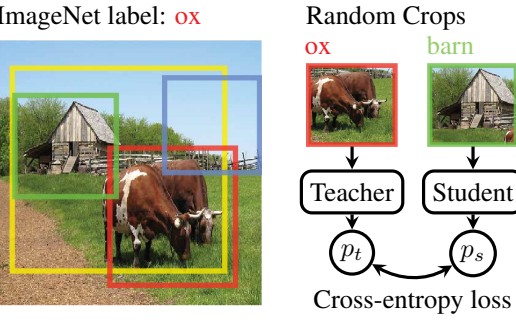

Figure 1: **(Left)** Multi-object example from ImageNet. Taken from (Yun et al., 2021). **(Right)**.

ever, does not always hold: In standard practice, the student and teacher networks see two random augmentations of an image and these augmentations possibly change the input content, breaking the assumption. As illustrated in Fig. 1, this issue is common even in ImageNet-1K (IN1K) (Deng et al., 2009), where roughly 20% of images contain multiple classes (Tsipras et al., 2020).

To address this challenge, we propose using readily available or foundation model generated object localization in the form of bounding boxes. We treat object bounding boxes as weak-supervision and introduce **O**bject-level Self-**Dist**illation (ODIS), a pretraining method that shifts self-distillation granularity from whole images to individual objects (see Fig. 2). This decomposition turns a complex scene-level task into simpler object-centric subproblems. ODIS focuses computation on semantically meaningful content via ① **object-aware cropping**, which ensures the teacher and student view different augmentations of the *same* object, and ② **masked attention**, which guides the learning objective toward object-centric representations useful for downstream tasks such as classification. We empirically demonstrate that these two innovations exploit the bounding boxes more efficiently than alternative approaches, such as cropping objects.

While these ideas can also be incorporated into contrastive learning (Chen et al., 2020), masked image modeling (He et al., 2022), and multi-modal frameworks (Radford et al., 2021), we focus on object-level self-distillation. As in image-level self-distillation Caron et al. (2021); Zhou et al. (2021); Oquab et al. (2023), we do not use object labels. Yet, unlike prior work, we perform self-distillation at the object level.

Across all image- and patch-level benchmarks we evaluate, ODIS consistently outperforms state-of-the-art image-level distillation baselines trained at the same scale. Beyond image-level classification, ODIS also improves patch-level performance on in-context scene-understanding and linear segmentation tasks (Balazevic et al., 2024; Lebailly et al., 2023). The resulting backbone transfers better to downstream classification datasets than our main baseline, iBOT Zhou et al. (2021). We also show how to use bounding boxes at inference to enable object-centric predictions: a ViT-Large pretrained with ODIS achieves $82.6\%$ $k$-NN accuracy on IN1K when bounding boxes are provided at test time.

Our main assumption is that bounding boxes are available during pretraining. In practice, this can be satisfied either by using datasets that include them or by applying off-the-shelf box extractors, which continue to improve. To test robustness of our approach, we also extract boxes from IN1K using two such extractors and find that ODIS improves over state-of-the-art baselines with either one. This aligns with recent efforts in data filtering using pretrained models (Dehghani et al., 2023; Oquab et al., 2023; Siméoni et al., 2025; Vo et al., 2024), which highlight the value of integrating pretrained models into learning pipelines to obtain stronger backbones.

## 2 RELATED WORK

**Image-level pretraining.** Inspired by large-scale NLP pretraining, many vision works adopt similar strategies. Early methods use tasks such as reconstructing masked patches (He et al., 2022; Bao et al., 2021), including variants that operate in feature space (Assran et al., 2023). These approaches improve performance when fine-tuned on downstream tasks.

However, our focus is on representations that work well without extra fine-tuning, which brings us closer to discriminative image-level self-distillation (Grill et al., 2020; Caron et al., 2021; Zhou et al., 2021; Oquab et al., 2023). These methods use a teacher-student framework (Tarvainen & Valpola, 2017) and avoid the negative pairs used in contrastive learning (Chen et al., 2020). iBOT (Zhou et al., 2021) combines image-level self-distillation (Caron et al., 2021) with a patch-level loss inspired by masked language modeling (Devlin et al., 2018). Building on this, Oquab et al. (2023) add algorithmic improvements for stable large-scale training and scale ViTs to a 142M-image dataset and a 1B-parameter model, reaching state-of-the-art results on diverse tasks. Our work also builds on iBOT (Zhou et al., 2021), however we focus on enhancing the learning objective to a finer object-level granularity instead of scaling the pretraining.

**Object-level pretraining.** A parallel line of research explores finer-grained pretraining objectives, ranging from pixel level distillation (O Pinheiro et al., 2020) to patch-level (Wang et al., 2021) and full object-level formulations (Hénaff et al., 2021; 2022; Xie et al., 2021; Stegmüller et al., 2023; Wen et al., 2022). These works primarily target dense tasks, such as object detection and semantic segmentation, defined at the pixel or patch level rather than at the image level. They are evaluated on

these tasks either with full fine-tuning (Hénaff et al., 2021; 2022; Wen et al., 2022) or with a linear head (Xie et al., 2021; Stegmüller et al., 2023). Object-centric alignment has also been studied in vision-language models, where RegionCLIP (Zhong et al., 2022) extends CLIP (Radford et al., 2021) by aligning region embeddings with text. Instead, we focus on pretraining on images solely.

Closest to our approach are (Hénaff et al., 2021; 2022). Of particular interest, Hénaff et al. (2021) proposes an object-level contrastive loss using segmentation masks, but forms object embeddings with average (linear) pooling over dense features, which limits expressivity. In contrast, our masked attention mechanism uses bounding boxes at each transformer layer, building highly nonlinear object-level representations. More importantly, while prior work emphasizes fine-tuning for object detection and segmentation, our goal is to learn general-purpose object-level representations useful for downstream tasks out of the box.

**Object-centric learning methods.**   Object-centric learning (OCL) (Burgess et al., 2019; Locatello et al., 2020; Seitzer et al., 2022; Didolkar et al., 2025) aims to discover object-like structure and is usually evaluated by unsupervised object segmentation. Yet, modern segmentation foundation models (Kirillov et al., 2023) outperform current OCL methods in zero-shot scenarios (Rubinstein et al., 2025), raising questions about whether OCL is useful for broad vision tasks. Object-centric representations are further assumed to capture compositional structures useful for visual reasoning tasks (Ding et al., 2021; Mamaghan et al., 2024), however it remains unclear how transferable their learned object-level features are beyond visual reasoning and segmentation, since the quality of their learned representations is not tested on standard benchmarks.

In contrast, we focus on object-level representations that work directly on standard benchmarks such as ImageNet $k$-NN. By coupling object-level distillation with bounding boxes, we connect insights from OCL and large-scale self-distillation, and we expect the resulting representations to also benefit OCL-related tasks.

# 3 PRELIMINARIES

In this section, we briefly review the self-distillation algorithms of DINO (Caron et al., 2021) and iBOT (Zhou et al., 2021), as our method builds on them.

**Input.**   An input image $x \in \mathbb{R}^{C \times H_{\text{img}} \times W_{\text{img}}}$ is transformed via standard augmentations such as random cropping followed by a resize in order to obtain two random global views: $x^{(1)}, x^{(2)} \in \mathbb{R}^{C \times H_{\text{resize}} \times W_{\text{resize}}}$[1]. Two views $x^{(1)}$ and $x^{(2)}$ are divided into $H \times W$ patches and linearly projected to a $D$ dimensional embedding space: $\tilde{x}^{(1)}, \tilde{x}^{(2)} \in \mathbb{R}^{(HW) \times D}$. State-of-the-art pretraining approaches (Caron et al., 2021; Zhou et al., 2021; Oquab et al., 2023) typically concatenate the $[\text{CLS}] \in \mathbb{R}^{1 \times D}$ token which summarizes the image-level visual information: $[[\text{CLS}], \tilde{x}] \in \mathbb{R}^{(1+HW) \times D}$.

**Network architecture.**   The algorithm is implemented using a pair of student and teacher networks: $g_s = h_s \circ b_s$ and $g_t = h_t \circ b_t$, with ViT backbones $b_s, b_t$ and the MLP prediction heads $h_s, h_t$. The output activation of the MLP prediction heads $h_s, h_t$ are softmax with temperatures $t_s > t_t$.

**Visual representations.**   Visual representations are the outputs of the ViT backbones. Although both the teacher and student process both global views in practice, for clarity we illustrate a simplified scenario where the teacher receives `view 1` and the student receives `view 2` (using the view color coding in Fig. 2b).

$$z_{[\text{CLS}],t}^{(1)}, z_{\text{patches},t}^{(1)} = b_t([[\text{CLS}]^{(1)}, \tilde{x}^{(1)}]), \qquad \text{teacher} - \text{view 1} \qquad (1)$$

$$z_{[\text{CLS}],s}^{(2)}, z_{\text{patches},s}^{(2)} = b_s([[\text{CLS}]^{(2)}, \tilde{x}^{(2)}]), \qquad \text{student} - \text{view 2} \qquad (2)$$

with image-level representation $z_{[\text{CLS}]} \in \mathbb{R}^{1 \times D}$ and patch-level representations $z_{\text{patches}} \in \mathbb{R}^{HW \times D}$.

**Image-level objective (DINO Loss) (Caron et al., 2021).**   MLP heads take the representations $z_{[\text{CLS}]}$ as input and produce probability vectors $p_s, p_t$, e.g., $p_{[\text{CLS}],s} = h_s(z_{[\text{CLS}]})$. We take `CrossEntropy` (CE) loss between probability vectors $p_s, p_t$ that correspond to distinct views

---

[1]For simplicity, we ignore local crops for now.

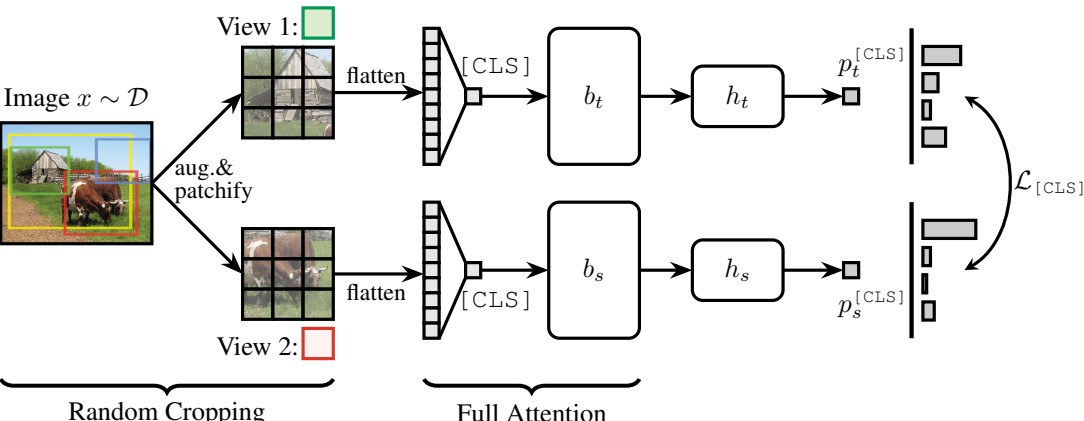

(a) Image-level self-distillation via [CLS] token with Random Cropping and Full Attention.

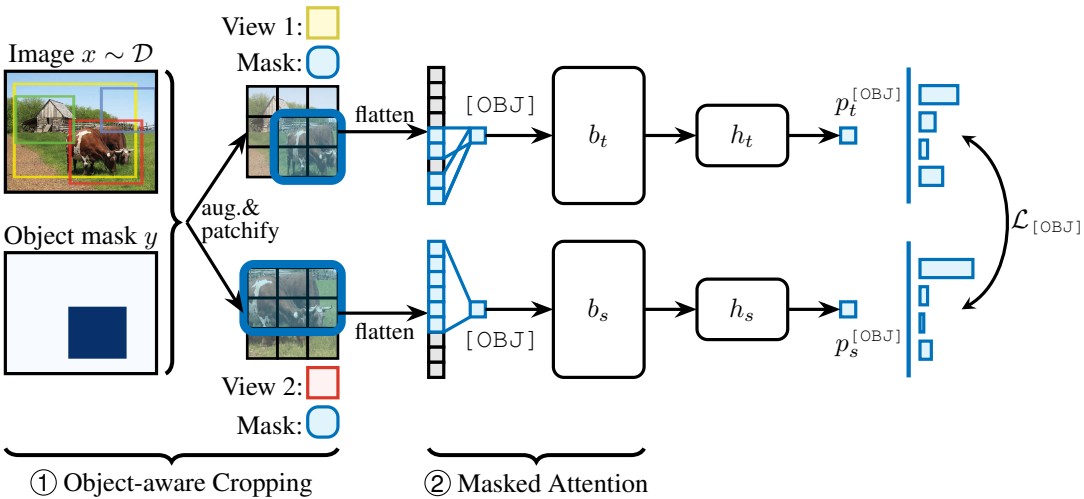

(b) Object-level self-distillation via [OBJ] token with Object-aware Cropping and Masked Attention.

Figure 2: **Image-level vs. Object-level distillation.** **(a)** Standard random cropping have no inherent mechanism to ensure that the student and teacher receive the same object as input. Hence, the distilled [CLS] tokens may summarize semantically different entities. **(b)** Our approach resolves this issue by ① Object-aware Cropping that uses object masks. Further, ② Masked Attention guides the [OBJ] token to pool information only from object tokens, leading to better representations.

$x^{(1)}, x^{(2)2}$:

$$p^{(1)}_{[CLS],t} = h_t(z^{(1)}_{[CLS]}), \qquad\qquad \texttt{teacher - view 1 [CLS]} \quad (3)$$

$$p^{(2)}_{[CLS],s} = h_s(z^{(2)}_{[CLS]}), \qquad\qquad \texttt{student - view 2 [CLS]} \quad (4)$$

$$\mathcal{L}_{[CLS]} = \texttt{CrossEntropy}(p^{(1)}_{[CLS],t}, p^{(2)}_{[CLS],s}), \qquad\qquad \texttt{DINO loss} \quad (5)$$

For clarity, we only provided the loss term for the simplified scenario above. The full loss is symmetric across views: $\mathcal{L}_{[CLS]} = \frac{1}{2}\big(\texttt{CE}(p^{(1)}_{[CLS],t}, p^{(2)}_{[CLS],s}) + \texttt{CE}(p^{(2)}_{[CLS],t}, p^{(1)}_{[CLS],s})\big)$.

**Patch-level objective and iBOT loss (Zhou et al., 2021).** iBOT creates an additional masked-image modeling task. For the student network input, it applies a random binary mask $m_1 \in \{0,1\}^{HW}$ to the input patch tokens $\tilde{x}^{(1)}, \tilde{x}^{(2)} \in \mathbb{R}^{(HW) \times D}$. The masking replaces corresponding tokens by a general [PATCH] token, e.g., $\tilde{x}^{(1)}[m_1] := [PATCH]^3$. The teacher receives unmasked

---

[2]Cross-entropy is defined as the dot product: $\texttt{CrossEntropy}(\mathbf{a}, \mathbf{b}) = \sum_i a_i \log b_i$.

[3]We use $\texttt{torch}$ boolean mask notation in $\tilde{x}^{(1)}[m_1]$, selecting entries $i \in [HW]$ in $\tilde{x}^{(1)}[i]$ when $m_1[i] = 1$.

patch tokens. Similar to image-level loss, MLP prediction heads produce probability vectors, e.g., $p_{\text{patches},s} = h_s(z_{\text{patches},s})$. In contrast to the cross-view formulation of the image-level loss, the patch-level loss is computed as follows for a single patch with patch index $i \in [HW]$ corresponding to the same views:

$$p^{(1)}_{\text{patches},t} = h_t(z^{(1)}_{\text{patches}}), \qquad\qquad \text{teacher} - \text{view 1 unmask} \quad (6)$$

$$p^{(1)}_{\text{patches},s} = h_s(z^{(1)}_{\text{patches}}), \qquad\qquad \text{student} - \text{view 1 mask} \quad (7)$$

$$\mathcal{L}_{\text{[PATCH]}}[i] = m_1[i]\,\text{CrossEntropy}(p^{(1)}_{\text{patches},t}[i], p^{(1)}_{\text{patches},s}[i]) \quad \text{patch loss for } i \in [HW] \quad (8)$$

which is summed over all masked patches: $\mathcal{L}_{\text{[PATCH]}} = -\frac{1}{\sum_j m(j)} \sum_{i \in [HW]} \mathcal{L}_{\text{[PATCH]}}[i]$. The iBOT loss sums up image- and patch-level losses: $\mathcal{L}_{\text{iBOT}} = \mathcal{L}_{\text{[CLS]}} + \mathcal{L}_{\text{[PATCH]}}$.

**Optimization.** The student network parameters $\theta_s$ are updated at every step via stochastic gradient descent. The gradients do not flow back to the teacher network, instead the teacher parameters $\theta_t$ are updated at every epoch as an exponential moving average (EMA) of the student parameters $\theta_s$: $\theta_t = \lambda\theta_t + (1 - \lambda)\theta_s$ (Tarvainen & Valpola, 2017).

## 4 OBJECT-LEVEL SELF-DISTILLATION

Next, we detail **O**bject-level Self-**Dis**tillation (ODIS), our proposed pretraining method that redefines self-distillation at the object level rather than the conventional image level. ODIS is built around two key components: ① object-aware cropping, which ensures that both student and teacher networks receive distinct views of the same object, and ② masked attention, which focuses the learning objective on objects, illustrated in Figs. 2 and 3. Together, these components guide the model toward learning richer, object-centric representations that transfer effectively to downstream tasks such as classification.

① **Object-aware cropping.** In addition to an input image $x \in \mathbb{R}^{C \times H_{\text{img}} \times W_{\text{img}}}$, the model also receives a bounding box $y$ (more specifically, the coordinates of the top-left and bottom-right corners of the encapsulating box). While augmenting the input image $x$ to obtain two random views $x^{(1)}, x^{(2)} \in \mathbb{R}^{C \times H_{\text{resize}} \times W_{\text{resize}}}$, we apply the same spatial transformations to the input box to obtain two boxes $y^{(1)}, y^{(2)}$ aligned with the image views. Similar to image views, the box views are patchified, but into binary matrices $\tilde{y}^{(1)}, \tilde{y}^{(2)} \in \{0, 1\}^{HW}$ where 1 denotes a patch falling inside the bounding box. We ensure that the target object is present in both global views by randomly cropping up to 20 times and keeping the global views that contain the target object.

When the input includes multiple distinct objects during training, we sample a single target object and the corresponding box per forward pass. To sample the target object, we consider two object sampling strategies: at random or at random proportional to object areas (see ablations for details). This way, the model targets a single object per forward pass while being able to see all objects in an image throughout training epochs.

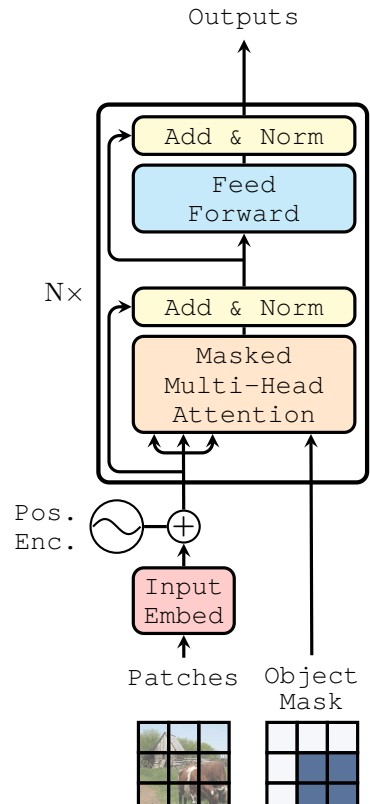

Figure 3: **Masked Attention with Object Bounding Boxes.**

② **Masked attention.** Previous works (Caron et al., 2021; Zhou et al., 2021; Oquab et al., 2023) concatenate the patch sequence with the global [CLS] token, summarizing image-level information. Similarly, we introduce a new *object-level token* [OBJ] $\in \mathbb{R}^{1 \times D}$ that represents only the features of the target object. Using masked attention, [OBJ] only attends to those patches where the object is present based on the object bounding box views $\tilde{y}^{(1)}, \tilde{y}^{(2)}$. Again, we use the scenario where the teacher network takes view 1 as input and the student network takes view 2:

$$z^{(1)}_{\text{[OBJ]},t}, z^{(1)}_{\text{patches},t} = b_t([\,[\text{OBJ}]^{(1)}, \tilde{x}^{(1)}], \text{obj-attn-mask} = \tilde{y}^{(1)}), \quad \text{teacher} - \text{view 1} \quad (9)$$

$$z^{(2)}_{\text{[OBJ]},s}, z^{(2)}_{\text{patches},s} = b_s([\,[\text{OBJ}]^{(2)}, \tilde{x}^{(2)}], \text{obj-attn-mask} = \tilde{y}^{(2)}), \quad \text{student} - \text{view 2} \quad (10)$$

Notice that each transformer layer uses the object bounding box as input to the `MaskedMultiHeadAttention` (`MaskedMHA`) block as in Fig. 3 to update the attention scores of the `[OBJ]` token. This leads to object-level representations that are highly nonlinear mixtures of the corresponding patch tokens, as opposed to works that consider average pooling of the patches (Hénaff et al., 2021; 2022; Lebailly et al., 2023).

In standard ViTs, `[CLS]` token can attend to any other token, including large, textured, or crop-overlapping background patches. Our masked-attention design allows `[OBJ]` token to *pool* exclusively from tokens that fall inside the bounding box, thereby yielding a cleaner object representation with a higher signal-to-noise ratio. Importantly, the restriction applies *only* to the `[OBJ]` token, i.e., the patch tokens belonging to the object still participate in full, unmasked self-attention with the rest of the patches in the image. Thus they can pull in whatever context is genuinely informative. For example, barn walls and grass texture in Fig. 1 carry information about the cow tokens; hence, they may help object tokens to better describe the object.

**Object-level objective.** MLP prediction heads take the representations $z_{\texttt{[OBJ]}}$ as input and produce probability vectors $p_s, p_t$, e.g., $p_{\texttt{[OBJ]},s} = h_s(z_{\texttt{[OBJ]}})$. For clarity, we again provide a simplistic example computing the cross-entropy loss only in one direction. We take cross-entropy loss between probability vectors $p_s, p_t$ that correspond to distinct views $x^{(1)}, x^{(2)}$:

$$p_{\texttt{[OBJ]},t}^{(1)} = h_t(z_{\texttt{[OBJ]}}^{(1)}), \qquad\qquad \texttt{teacher - view 1 [OBJ]} \qquad (11)$$

$$p_{\texttt{[OBJ]},s}^{(2)} = h_s(z_{\texttt{[OBJ]}}^{(2)}), \qquad\qquad \texttt{student - view 2 [OBJ]} \qquad (12)$$

$$\mathcal{L}_{\texttt{[OBJ]}} = \texttt{CrossEntropy}(p_{\texttt{[OBJ]},t}^{(1)}, p_{\texttt{[OBJ]},s}^{(2)}) \qquad \texttt{object-level loss} \qquad (13)$$

while the loss is symmetric: $\mathcal{L}_{\texttt{[OBJ]}} = \frac{1}{2}(\texttt{CE}(p_{\texttt{[OBJ]},t}^{(1)}, p_{\texttt{[OBJ]},s}^{(2)}) + \texttt{CE}(p_{\texttt{[OBJ]},t}^{(2)}, p_{\texttt{[OBJ]},s}^{(1)}))$.

**Final loss.** Our final loss sums the object-level loss with the patch-level loss described in Section 3:

$$\mathcal{L}_{\text{ODIS}} = \mathcal{L}_{\texttt{[OBJ]}} + \mathcal{L}_{\texttt{[PATCH]}}. \qquad (14)$$

As the patch-level masking strategy, we use random block masking as in (Zhou et al., 2021).

**Discussion on the use of object bounding boxes.** Modern pretraining approaches adopt weak supervision signals such as paired text (Radford et al., 2021; Tschannen et al., 2025), yet it still overlooks the simplest one: the bounding boxes already bundled with datasets such as ImageNet-1k and COCO. In ODIS we treat these masks as free supervision, feeding object-aware crops during pre-training for the network to learn spatially grounded features. For pretraining datasets without bounding boxes, we propose to run off-the-shelf bounding box extractors. Our experiments show that ODIS pretrained with bounding boxes from off-the-shelf extractors still yields improvements over not using any boxes in pretraining. Our method does not require object annotations at inference time. Yet, as we show in our experiments they enhance performance of all methods considered if these annotations are available in inference time. Our findings suggest that pretraining pipelines should default to using bounding boxes whenever they are available or can be generated automatically.

**Implementation details.** We follow the ViT architectures and the pretraining setups in previous works (Caron et al., 2021; Zhou et al., 2021), as further detailed in Appendix B. We use ViTs of different sizes, ViT-Small/16, ViT-Base/16 and ViT-Large/16 with patch size equal to 16. We pretrain our model separately on COCO and IN1K. In COCO, each image has on average $\sim 7$ distinct object instances of $\sim 150$ object classes. We convert the provided segmentation masks to bounding boxes. For IN1k, a single object box is provided for each image, locating the main object. All 50k validation images in IN1K have a valid box, while only $500k / 1.2M$ training images have one. For the images missing the box, we assume that the main object covers the whole image. On IN1k, we also ablate different object boxes produced by off-the-shelf bounding box extraction models: YOLO (Redmon et al., 2016) and a multi-modal ViT (Maaz et al., 2022, MAVL) provide class-agnostic boxes, possibly with multiple distinct objects for each image.

## 5 EXPERIMENTS

We evaluate ODIS through a comprehensive set of experiments to assess its effectiveness at both image- and patch-level representation learning. Our results show that (i) incorporating ground-truth

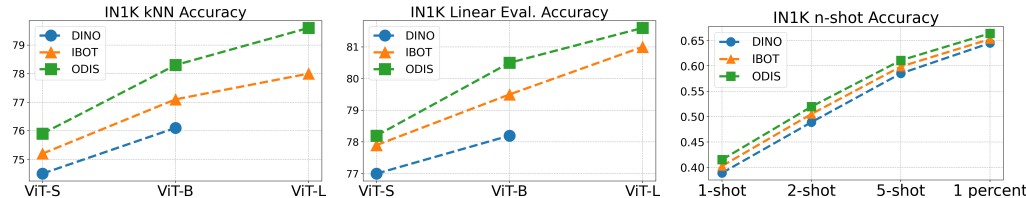

Figure 4: **ODIS yields superior image-level representations.** We report $k$-NN, linear probing, and few-shot classification accuracy on IN1K. $k$-NN and linear probing results for DINO and iBOT are taken from their result tables. Few-shot classification is run using their official checkpoints.

bounding boxes during pretraining consistently improves image- and patch-level representations (Secs. 5.1, 5.4); (ii) the resulting backbone transfers more effectively to diverse, out-of-distribution datasets (Sec. 5.2); (iii) pretraining with bounding boxes automatically extracted by off-the-shelf detectors still yields clear gains over strong self-supervised baselines (Sec. 5.3). We further show that using object-level annotations at inference time alleviates the single-label limitations of IN1K and further improves evaluation accuracy in App. D.1. We provide experimental details in App. C and ablations in App. E.

## 5.1 INCORPORATING BOUNDING BOXES IN PRETRAINING YIELDS SUPERIOR IMAGE-LEVEL REPRESENTATIONS

We first study how incorporating object-level annotations *during pretraining* improves learning signal, and hence yields better visual features. To measure the quality of image representations, we follow previous works (Caron et al., 2021; Zhou et al., 2021; Assran et al., 2022) and evaluate our pretrained models on a standard vision benchmark: *classification on IN1K test set*. We focus on off-the-shelf visual feature quality. Hence, we consider simple classifiers on frozen features: $k$-NN and linear classifiers, including $n$-shot scenarios. We compare our models with strong self-distillation baselines that are at the same scale: DINO (Caron et al., 2021) and iBOT (Zhou et al., 2021).

We pretrain our models on two different datasets with different characteristics: IN1K (Deng et al., 2009), that contains object-centric images, and COCO (Lin et al., 2014), that contains scene-centric, multi-object images. During pretraining, we use readily-available bounding boxes of both datasets.

**Pretraining on object-centric data (IN1K).** On IN1K, we train three model sizes: ViT-S, ViT-B and ViT-L. We follow pretraining and evaluation protocols in Caron et al. (2021); Zhou et al. (2021). We further consider a few-shot learning setup following Assran et al. (2022), where a linear classifier is trained using 1, 2, or 5 labeled images or 1% of the training data (about ∼13 images per class). We show results in Fig. 4. ODIS outperforms the baselines in all evaluation setups, implying our backbone has learned richer in-domain representations. In $k$-NN classification, ODIS provides significant performance improvements compared to our main baseline iBOT: +0.7 for ViT-S, +1.2 for ViT-B, and +1.6 for ViT-L. We emphasize that these are significant performance improvements, especially, considering that iBOT improves +0.7 and +1.0 over DINO for ViT-S and ViT-B. We also see that scaling the model boosts the gains.

**Pretraining on scene-centric data (COCO).** Real-world images often depict scenes with several objects/entities, which are much more complex than carefully-curated, object-centric IN1K dataset. While pretraining on such uncurated data reduces the performance (Siméoni et al., 2025), our approach, which utilizes bounding boxes for distillation and attention, is expected to be more robust. To validate this hypothesis, we train our model on COCO dataset. We pretrain a ViT-S model on the COCO dataset using DINO, iBOT, and ODIS objectives. We freeze the pretrained models, and build $k$-NN classifiers on frozen fea-

Table 1: $k$-**NN IN1K for scene-centric pretraining**. All model sizes are ViT-S.

| Model | Epochs | Pretrain. | $k$-NN |
|-------|--------|-----------|--------|
| DINO  | 300    | Coco      | 36.9   |
| iBOT  | 300    | Coco      | 41.8   |
| ODIS  | 300    | Coco      | **43.3** ↑1.5 |

Table 2: **Transfer learning experiment.** Across 10 different datasets, we train a linear head on top of iBOT and ODIS ViT-L backbones pretrained on IN1K. ODIS outperforms iBOT on 9/10 datasets.

| Model | Food | C10 | C100 | SUN | Cars | Aircr | DTD | Pets | Cal101 | Flowers | Avg. |
|---|---|---|---|---|---|---|---|---|---|---|---|
| iBOT | 84.8 | 97.7 | 88.3 | **68.0** | 74.6 | 67.0 | 77.7 | 93.4 | 95.5 | 97.3 | 84.4 |
| ODIS | **85.0** | **98.0** | **88.5** | 67.9 | **75.8** | **68.3** | **78.5** | **94.1** | **96.5** | **97.9** | **86.1** ↑1.7 |

tures for IN1K test set. We see a similar trend: the $k$-NN performance on IN1k (+1.5) improves significantly compared to iBOT.

**Conclusion**.  Using bounding boxes during pretraining, ODIS yields better image representations than strong baselines, such as DINO and iBOT, that do not use these annotations. This indicates that using bounding boxes in pretraining improves the learning signal, and object-level annotations should be used when they are readily-available. Our findings consistently hold across two pretraining datasets (IN1K and COCO), three model sizes (ViT-S, ViT-B, ViT-L), and using different metrics such as $k$-NN, linear, and $n$-shot accuracy on frozen features.

## 5.2 Our backbone generalizes better to out-of-distribution datasets

Next, we study how incorporating bounding boxes during pretraining affects transfer learning. For this, we transfer DINO, iBOT, and ODIS backbones pretrained on IN1K to two new tasks: classification on out-of-distribution datasets with linear probing (Chen et al., 2020) and image retrieval (Caron et al., 2021). We again focus on the quality of frozen visual features.

**Linear head on downstream datasets**   The first experiment follows Chen et al. (2020), where a linear classifier is trained on top of the frozen backbones of iBOT and ODIS ViT-L across 10 natural image datasets. We follow the evaluation protocol provided in Chen et al. (2020). As shown in Table 2, ODIS transfers better to 9 out of 10 datasets, resulting in a +1.7 average accuracy increase.

**Image retrieval**.   Our next task is image retrieval from the revisited Paris and Oxford datasets (Philbin et al., 2008; Radenović et al., 2018). We follow the evaluation pipeline in Caron et al. (2021) and use ViT-S. We retrieve images based on similarities of their frozen embeddings. We report the mean average precision (mAP) for the medium (M) and hard (H) splits. Table 3 shows that iBOT performs worse than DINO. We conjecture this is because iBOT's patch-level objective hurts the performance on this image-level task. Interestingly, our improved object-level distillation objective helps close the gap, making ODIS perform better than iBOT across all data splits and on par with DINO.

Table 3: **Image retrieval experiment**. We report the mean average precision (mAP) for the medium (M) and hard (H) splits.

| Model | Oxford | | Paris | |
|---|---|---|---|---|
| | M | H | M | H |
| DINO | **38.79** | **12.46** | 62.45 | 34.34 |
| iBOT | 37.34 | 10.22 | 61.08 | 34.72 |
| ODIS | 38.01 | 10.34 | **62.75** | **35.37** |

**Conclusion**.   Across twelve out-of-distribution datasets, ODIS produces better transferable image-level representations than our main baseline iBOT. Our findings show that using bounding boxes during pretraining improves transfer performance.

## 5.3 Auto-generated bounding boxes also yield consistent improvements

In the previous sections, we assumed that ground-truth bounding boxes are available during pretraining, an assumption may not apply to all pretraining datasets. Now, we study the case where bounding boxes are produced by an off-the-shelf extractor. Specifically, we consider YOLO (Redmon et al., 2016) and a multi-modal ViT (Maaz et al., 2022, MAVL). Neither of these tools was trained on IN1K, and they provide class-agnostic boxes, possibly with multiple distinct objects for each image. When an image contains multiple boxes, we sample a single one at random or proportional to box area for each forward pass. In practice, we found that sampling proportional to object area performed slightly better.

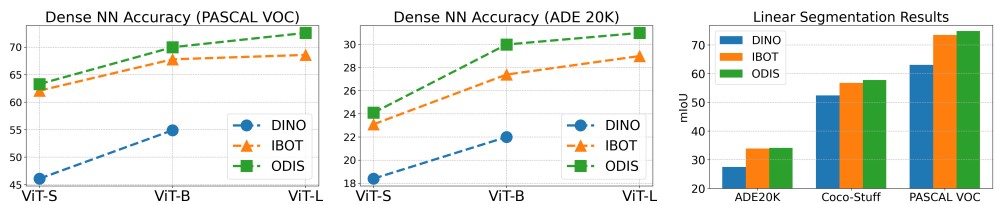

Figure 5: **ODIS yields superior patch-level representations**. mIoU results for dense nearest neighbor retrieval task: **(left)** PASCAL VOC and **(center)** ADE 20K. **(right)** mIoU results with a linear segmentation head across ADE20K, COCO-Stuff and PASCAL-VOC with model size ViT-B.

We ablate the source of bounding boxes by training ODIS using ground-truth, YOLO-, and MAVL-generated boxes. We compare these with iBOT. For all methods, we train a ViT-S for 150 epochs on IN1K, and measure $k$-NN accuracy on IN1K test set. We show results in Table 4.The results reveal that regardless of the source of bounding boxes, ODIS produces a higher $k$-NN accuracy than iBOT, while using the ground-truth boxes gives the best performance.

Table 4: Ablation on using different bounding boxes during pretraining.

| Model | Segmentor | $k$-NN |
|-------|-----------|--------|
| iBOT | - | 70.9 |
| ODIS | YOLO | 71.2 |
| ODIS | MAVL | 71.3 |
| ODIS | Ground-truth | 71.9 |

**Conclusion**. ODIS consistently outperforms training without boxes, even when the boxes are generated by off-the-shelf tools. This indicates our method is robust to imperfect or class-agnostic boxes, while access to higher-quality boxes yields the best performance.

### 5.4 INCORPORATING BOUNDING BOXES IN PRETRAINING ALSO YIELDS SUPERIOR PATCH-LEVEL REPRESENTATIONS

Finally, we shift our focus to patch-level representations, from image-level. We investigate whether our approach also improves patch-level representations on out-of-domain datasets: PASCAL VOC, ADE20K and Coco-Stuff. We follow previous works (Balazevic et al., 2024; Lebailly et al., 2023; Pariza et al., 2024), and choose two patch-level tasks that are designed for frozen features: dense nearest neighbor retrieval and linear segmentation. We report mean intersection over union (mIoU) for both tasks. Dense nearest neighbor retrieval is similar to $k$-NN classification, where each test patch is assigned a label as a weighted sum of the labels of its close training neighbor patches. We summarize findings in Fig. 5, and provide detailed results in Table 6, including comparisons with additional baselines.

For both tasks, we consistently see mIoU gains for ODIS patch representations compared to iBOT across all datasets and model sizes. In linear segmentation, ODIS slightly improves iBOT, while the gains are more substantial on dense nearest neighbor retrieval: On PASCAL VOC, performance gains compared to iBOT are (i) $+1.2$ for ViT-S, (ii) $+2.2$ for ViT-B and $+4.0$ for ViT-L. On ADE20k, performance gains compared to iBOT are (i) $+1.0$ for ViT-S, (ii) $+2.6$ for ViT-B and $+2.0$ for ViT-L.

**Conclusion.** In addition to superior image-level features, using object annotations during pretraining also yields better *patch-level* features, compared to not using these annotations.

## 6 CONCLUSION

In this work, we explore object-level self-distillation (ODIS) for pretraining vision foundation models. Our approach utilizes bounding boxes encapsulating objects in an image for object-aware cropping (to ensure that inputs to teacher and student contain the same object) and masked attention (to focus the learning signal on objects). We empirically demonstrate that ODIS learns superior image- and patch-level representations that also transfer better to new datasets. ODIS relies on bounding boxes during pretraining. This assumption can be relieved by exploiting off-the-shelf bounding box extractors as we showed that incorporating them improves the backbone over baselines such as iBOT. In future work, we plan to scale our method to larger model sizes and datasets.

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

## A    CONNECTIONS WITH OTHER MASKED MODELING FRAMEWORKS AND GRAPH LEARNING

**Connection to BERT, Masked Image Modeling, Masked Autoencoders**.   Masked image modeling with vision transformers draws inspiration from masked language modeling in NLP (Devlin et al., 2018), where masked words are predicted from their surrounding context. In vision, similar strategies have been applied: models predict masked image patches (He et al., 2022, Masked Autoencoders) or discrete visual tokens (Bao et al., 2021, BEiT) based on neighboring content, leading to highly effective generative frameworks. Self-supervised approaches such as iBOT (Zhou et al., 2021) and DINOv2 (Oquab et al., 2023) extend this idea using masked patch prediction combined with a distillation objective.

Despite their empirical success, these vision models diverge fundamentally from their textual counterparts: while language models predict meaningful and discrete units like words or subword tokens, masked vision models typically predict arbitrary patches, which are often unidentifiable parts of objects or even background. Moreover, whereas text tokenizers increasingly align with linguistic units (syllables or words), vision lacks such semantically grounded units. In this work, we address this gap by proposing objects, which are the natural semantic units of visual scenes, as prediction targets. Analogous to words in language, objects in images offer coherent, interpretable units for representation learning.

**Connection to Graph and Subgraph Pooling**.   We can view each image as a fully-connected graph, where nodes represent patches and node representations correspond to patch embeddings. In this view, image-level distillation via `[CLS]` token corresponds to pooling a graph-level representation from all nodes. This is a hard task to solve. Object-level distillation via `[OBJ]` token corresponds to pooling a subgraph-level representation where the subgraph is located via segmentation maps. This is a simpler sub-task, that is aligned better with cross-entropy loss for scene-centric images.

## B    IMPLEMENTATION DETAILS

**ViT.** We follow previous works (Caron et al., 2021; Zhou et al., 2021) and use vision transformers (Dosovitskiy et al., 2020) in different sizes ViT-Small/16, ViT-Base/16 and ViT-Large/16 as the visual backbone $b(\cdot)$ with patch size equal to 16. We build on the code base of iBOT (Zhou et al., 2021). As commonly done, we use 2 global crops of size $224 \times 224$ with 10 local crops of size $96 \times 96$. The teacher only processes 2 global crops as input, while the student processes all crops. We use shared MLP heads for predicting the image- and patch-level probability vectors, with output dimension 8192.

**Pretraining setup.** We pretrain our models on COCO (Lin et al., 2014) and ImageNet-1k (IN1k) (Deng et al., 2009). To keep our results comparable, we follow the training setups used for COCO in (Lebailly et al., 2023) and for IN1k in (Zhou et al., 2021). For the COCO dataset, we pretrain ViT-S/16 for 300 epochs. For the IN1k dataset, we pretrain ViT-S/16 for 800 epochs, ViT-B for 400 epochs and ViT-L for 250 epochs. We use random block masking that masks $p \sim \mathcal{U}[0.1, 0.5]$ of the patches for the $50\%$ of the global crops (Zhou et al., 2021).

# C    EXPERIMENTAL DETAILS

## C.1    TRANSFER LEARNING: LINEAR CLASSIFICATION

We use the setup proposed in Chen et al. (2020). As in the original paper, we use LBFGS optimizer, the same train/test splits, and mean accuracy or mean accuracy per class metrics for corresponding datasets (please see Sec B.8 of Chen et al. (2020) for all details).

## C.2    OFF-THE-SHELF BOUNDING BOX EXTRACTORS

**External bounding boxes.** We generate object bounding boxes using two modern segmentation models: YOLO (Redmon et al., 2016) and MAVL (Maaz et al., 2022). They are both trained on COCO dataset and provide multi-object, class-agnostic bounding boxes. We sample objects with probabilities proportional to their areas for each forward pass.

## C.3    DENSE NEAREST NEIGHBOR RETRIEVAL

This task extends the standard image-level SSL benchmark to patches (Balazevic et al., 2024).

**Task description.**    For the training set, each image is split into $HW$ patches, and patch-label pairs $(p_i, y_i)_{i=1}^{HWN_{\text{train}}}$ are recorded, where $y_i$ is obtained by average pooling the pixel labels within patch $p_i$. We encode each patch $p_i$ into a feature vector $k_i = b_t(p_i)$ using the frozen ViT backbone $b_t(\cdot)$, and store a subset of these feature-label pairs in a memory bank $\mathcal{M} = \{(k_i, y_i)\}$ with different subsampling factors $\{1, 8, 64, 128\}$.

At test time, for each query patch $p_j$ in the validation set, we:

1. encode $p_j$ to obtain $q_j = b_t(p_j)$,
2. compute similarities between $q_j$ and all features in $\mathcal{M}$ using cross-attention (softmax-normalized),
3. predict the patch label $\hat{y}_j$ by a weighted average of the top-$k$ matching labels in $\mathcal{M}$, where each label is weighted by its attention score.

The predicted labels $\hat{y}_j$ for all patches of a test image are concatenated and then upsampled to the original image size via bilinear interpolation, yielding a final segmentation map.

**Evaluation setup.**    Following Balazevic et al. (2024); Lebailly et al. (2023), we pretrain ODIS and iBOT on both a scene-centric dataset, COCO (118k images), and an object-centric dataset, IN1k (1.28M images). We fix the maximum memory bank size $|\mathcal{M}|$ to 10,240,000 and sweep $k \in \{30, 50\}$.

## C.4    LINEAR SEGMENTATION

We follow the evaluation setups used in (Lebailly et al., 2023; Pariza et al., 2024). We use publicly available code repository of Pariza et al. (2024) to run the evaluations for DINO, iBOT and ODIS. We use official checkpoints for DINO and iBOT.

# D    ADDITIONAL RESULTS

## D.1    USING OBJECT-LEVEL ANNOTATIONS AT INFERENCE TIME IMPROVES MODEL EVALUATION

In this section, we shift our focus to using object-level annotations *at inference time*. We study whether using them at inference time improves visual features, when these annotations are available.

**Issues with single-label assumption in IN1K**. Our motivation comes from Tsipras et al. (2020), who demonstrated that more than 20 percent of IN1K images contain objects from multiple classes.

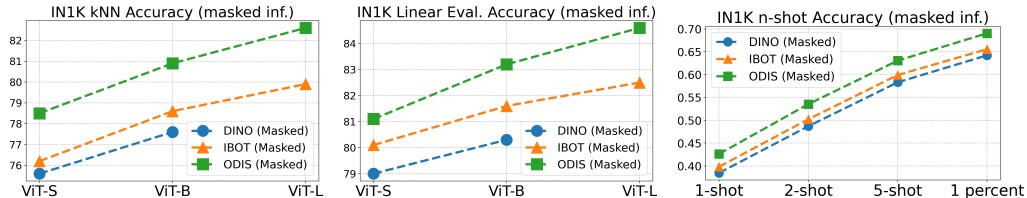

Figure 6: **Using bounding boxes in inference** for $k$-NN, linear probing, and few-shot classification accuracy on IN1K. To obtain "DINO/iBOT (Masked)" results, we use publicly available checkpoints.

Such examples might confound the previous analysis in Section 5.1 since the class of an input image is not always obvious. For example, Figs. 7 and 8 present examples in which NN retrieval based on `[CLS]` or `[OBJ]` token fails when the input image contains multiple potential target objects. In all cases, the retrieved image is semantically similar but labeled differently, leading to an incorrect match under the IN1K single-label protocol. Last-layer attention maps reveal that the model attends to *multiple* salient objects, highlighting that the embedding captures mixed object semantics. This entanglement undermines retrieval evaluation, especially when multiple plausible objects exist in the scene. To address this, we advocate the use of bounding boxes during inference to isolate individual object representations, enabling more faithful and interpretable evaluation of pretrained models.

**Our proposed remedy**. As IN1K test set comes with readily-available bounding boxes, we use IN1K test set as our benchmark. We consider utilizing bounding boxes in three different ways: (i) blacking out non-object patches in input images (Rubinstein et al., 2025), (ii) cropping out the object from input images and reshaping the resulting crop back to the image size, and (iii) *our approach*: masked attention where bounding boxes indicate which patches to mask.

**Results**. We summarize the masked attention results for each model in Fig. 6, and show detailed results for iBOT and ODIS including all box-usage approaches in Fig. 9. As expected, accuracies of all backbones increase thanks to our improved evaluation pipeline. Strikingly, for all metrics, the gap between ODIS and iBOT grows. For $k$-NN, the performance gains compared to iBOT are (i) $+2.3$ for ViT-S, (ii) $+2.3$ for ViT-B, and (iii) $+2.7$ for ViT-L (we note that iBOT improves over DINO by an average of only $+0.9$). Figure 10 shows a visual demonstration where even giving the object location as input to iBOT does not lead to correct classification, unlike ODIS.

While using masks via (ii) cropping or (iii) masked attention improves performance for all models, blacking out the background hurts performance. For iBOT, cropping provides the best results, closely followed by masked attention. For ODIS, masked-attention significantly increases the performance and provides the best results. This shows that the best approach to use boxes at inference time depends on whether the model used masked attention during training or not.

**Conclusion**. Using bounding boxes at inference time improves performance for all models considered. Therefore, we suggest using them whenever available for better model evaluation. Furthermore, ODIS benefits the most from using these boxes and enjoys significant performance improvements, since it learns to use bounding-box annotations during its pretraining.

# E ABLATIONS

Next, we list the findings of our ablation studies. We mainly ablate our method on COCO due to computational constraints, where pretrain a ViT-S for 300 epochs as in Lebailly et al. (2023). We report these results in Table 5. Additionally, we ablate using external masks for pretraining in IN1k and report the results in Table 4.

**Loss components.** We experimented with including an auxiliary image-level term $\mathcal{L}_i$ or not. Removing it improved patch-level accuracy and left object-level metrics more or less unchanged, so $\mathcal{L}_i$ is omitted in the final objective.

Table 5: **Effect of pretraining design choices.** We test object representations with $k$-NN on IN1k and test patch representations with mIoU on PASCAL VOC. PMLC: Patch masking for local crops. OALC: Object-aware local cropping. MALC: Masked-attention for local crops using object attention masks. 'Use Masks' and 'M.' refer to using the object segmentation masks at inference time for $k$-NN classification on IN1k.

| Model | Backbone | Epochs | Pretrain. | Use Masks | $k$-NN | mIoU |
|---|---|---|---|---|---|---|
| DINO | ViT-S | 300 | COCO | ✗ | 36.9 | 30.5 |
| iBOT | ViT-S | 300 | COCO | ✗ | 41.8 | 51.0 |
| iBOT+Masks | ViT-S | 300 | COCO | ✓ | 43.9 | 51.0 |
| *Loss components* | | | | | | |
| ODIS + Masks + $\mathcal{L}_i$ | ViT-S | 300 | COCO | ✓ | 44.5 | 51.0 |
| ODIS + Masks | ViT-S | 300 | COCO | ✓ | 46.0 | 54.9 |
| *Local Crop Configuration* | | | | | | |
| ODIS+PMLC+OALC+MALC | ViT-S | 300 | COCO | ✗ | 39.1 | 54.8 |
| +Masks | ViT-S | 300 | COCO | ✓ | 40.1 | 54.8 |
| - PMLC | ViT-S | 300 | COCO | ✓ | 41.4 | 54.0 |
| - OALC | ViT-S | 300 | COCO | ✓ | 42.6 | 54.6 |
| - MALC (=ODIS+Masks) | ViT-S | 300 | COCO | ✓ | 46.0 | 54.9 |
| *Object Sampling* | | | | | | |
| ODIS+Masks+ random sampl. | ViT-S | 300 | COCO | ✓ | 45.3 | 54.9 |
| ODIS+Masks+ random area sampl. | ViT-S | 300 | COCO | ✓ | 46.0 | 54.9 |

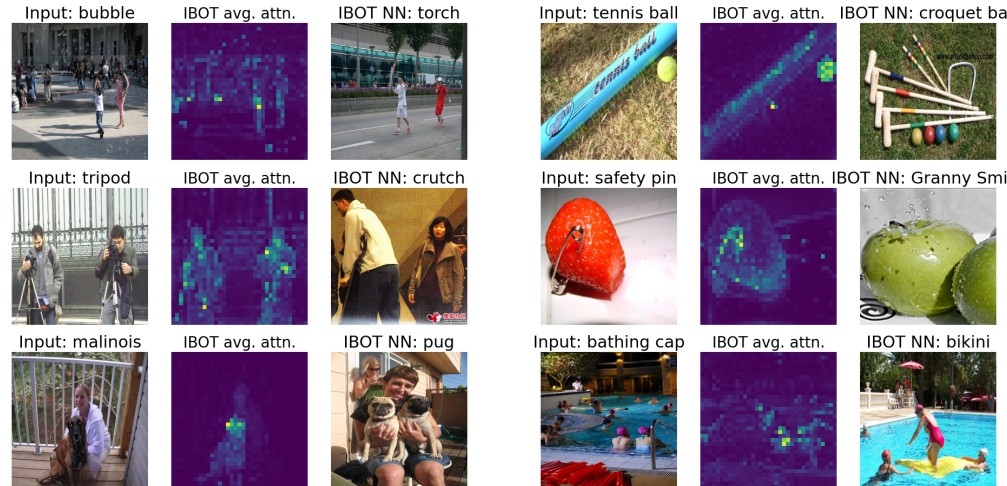

Figure 7: Examples showing how iBOT fails in retrieving a nearest neighbor with the correct class label in the presence of multiple objects. We propose to resolve this by using segmentation masks that specify the target object of interest.

**Local-crop configuration.** The best object representations arise when (i) tokens from local crops attend to all crop patches and (ii) the crops themselves are drawn from general, context-rich regions rather than object-aware windows.

**Object sampling.** On COCO, sampling objects with probability proportional to their area yields a small but consistent advantage over uniform sampling on object-level evaluations with a similar performance on patch-level evaluations.

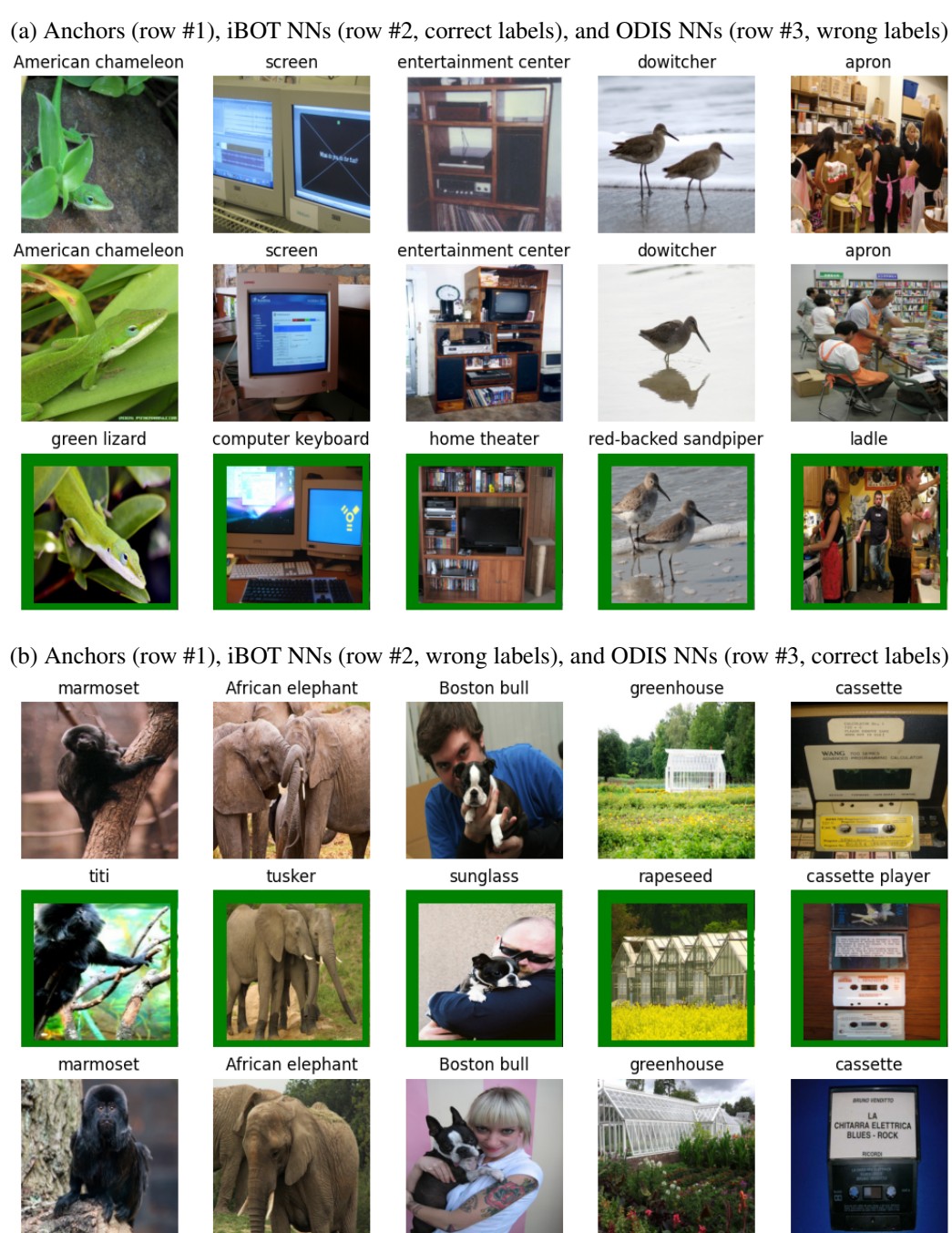

Figure 8: Qualitative examples showing the label noise in the nearest neighbor retrieval task. In the upper/lower panels, nearest neighbors retrieved by ODIS/iBOT have incorrect labels, respectively. For all retrieved items, a SOTA vision-text model (Tschannen et al., 2025, Siglip-2) queried with the input image and two potential labels decides that the retrieved item, which has an originally incorrect label, is actually of the same class as the anchor. While some images are clearly mislabeled during IN1K data curation, we also notice some images with multiple labels: computer keyboard and home theater in (a); sunglass and rapeseed in (b).

Table 6: **Dense nearest neighbor retrieval task**. We predict in-context segmentation labels and report mIoU. The models are pretrained on a *Scene-centric* dataset, COCO, or an *Object-centric* dataset, IN1k. The models are divided into two groups: (i) *Patch-level* group contains Hummingbird and CRIBO whose objectives primarily focus on increasing cross-image patch-level correspondence, specialized for the dense nearest neighbor retrieval task, (ii) *Higher-level* group contains MAE, DINO, iBOT and ODIS whose objectives focus on image- or object-level representations.

| Model | Back. | #Par. | Pretrain. | Epochs | PASCAL VOC | | | | ADE 20k | | | |
|---|---|---|---|---|---|---|---|---|---|---|---|---|
| | | | | | 1/128 | 1/64 | 1/8 | 1/1 | 1/128 | 1/64 | 1/8 | 1/1 |
| *Patch-lvl* | | | *Scene-c.* | | | | | | | | | |
| CRIBO | ViT-S | 21M | Coco | 300 | 39.1 | 44.0 | 52.8 | 58.1 | 10.9 | 12.8 | 18.4 | 23.4 |
| *Higher-lvl* | | | | | | | | | | | | |
| DINO | ViT-S | 21M | Coco | 300 | 16.2 | 18.4 | 25.5 | 31.9 | 6.1 | 6.9 | 9.7 | 13.0 |
| MAE | ViT-S | 21M | Coco | 300 | 8.5 | 9.3 | 12.2 | 15.9 | 3.7 | 4.1 | 5.4 | 6.8 |
| iBOT | ViT-S | 21M | Coco | 300 | 37.3 | 39.5 | 47.3 | 54.7 | 10.2 | 12.2 | 16.7 | 21.3 |
| ODIS | ViT-S | 21M | Coco | 300 | 42.7 | 43.6 | 51.8 | 57.7 ↑ 3.0 | 11.2 | 13.1 | 17.7 | 22.4 ↑ 1.1 |
| *Patch-lvl* | | | *Object-c.* | | | | | | | | | |
| CRIBO | ViT-S | 21M | IN1K | 800 | 52.7 | 59.3 | 69.3 | 73.2 | 13.7 | 16.5 | 23.2 | 28.3 |
| *Higher-lvl* | | | | | | | | | | | | |
| DINO | ViT-S | 21M | IN1K | 800 | 24.5 | 28.7 | 38.7 | 46.1 | 9.4 | 10.6 | 14.6 | 18.4 |
| iBOT | ViT-S | 21M | IN1K | 800 | 34.6 | 41.1 | 54.7 | 62.1 | 11.9 | 13.9 | 18.8 | 23.1 |
| ODIS | ViT-S | 21M | IN1K | 800 | 35.5 | 41.6 | 55.6 | 63.3 ↑ 1.2 | 12.1 | 14.2 | 19.3 | 24.1 ↑ 1.0 |
| *Patch-lvl* | | | | | | | | | | | | |
| Humming. | ViT-B | 85M | IN1K | 300 | 50.5 | 57.2 | - | 70.5 | 11.7 | 15.1 | - | 28.3 |
| CRIBO | ViT-B | 85M | IN1K | 400 | 50.5 | 60.3 | 70.8 | 74.9 | 13.2 | 16.5 | 23.6 | 30.0 |
| *Higher-lvl* | | | | | | | | | | | | |
| DINO | ViT-B | 85M | IN1K | 400 | 29.2 | 34.7 | 47.2 | 54.9 | 11.1 | 12.6 | 17.6 | 22.0 |
| MAE | ViT-B | 85M | IN1K | 1600 | 6.0 | 6.5 | 8.9 | 13.8 | 2.7 | 3.0 | 4.0 | 5.3 |
| iBOT | ViT-B | 85M | IN1K | 400 | 41.1 | 47.4 | 60.6 | 67.8 | 14.8 | 17.1 | 22.9 | 27.4 |
| ODIS | ViT-B | 85M | IN1K | 400 | 43.1 | 49.7 | 63.1 | 70.0 ↑ 2.2 | 16.2 | 18.8 | 25.1 | 30.0 ↑ 2.6 |
| iBOT | ViT-L | 307M | IN1K | 250 | 41.1 | 46.7 | 60.8 | 68.6 | 15.8 | 18.3 | 24.4 | 29.0 |
| ODIS | ViT-L | 307M | IN1K | 250 | 44.6 | 51.2 | 65.4 | 72.6 ↑ 4.0 | 17.1 | 19.7 | 26.1 | 31.0 ↑ 2.0 |

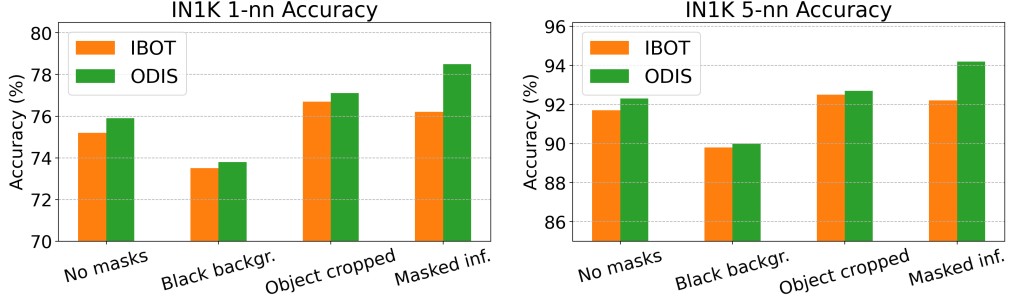

Figure 9: $k$-nn accuracies in different inference settings. The panels show the top-1 and top-5 NN accuracies as proposed by Tsipras et al. (2020) to tackle the multi-label nature of IN1K. In addition to the standard setting without any masks, we experiment with three ways of exploiting bounding boxes: (i) blacking out the area outside of the box, (ii) cropping the object inside the box and reshaping to the image size, and (iii) our proposed masked inference. Across both metrics, masked inference with ODIS significantly outperforms the alternatives. For iBOT, object cropping seems to perform slightly better than masked inference.

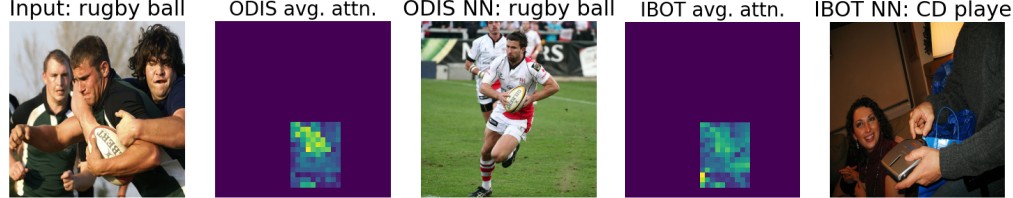

Figure 10: An example input, ODIS and iBOT attention maps using inference-time masks, and retrieved nearest neighbors. Without object masks, the class identity is unclear. Despite using the object mask, iBOT mistakenly attends to the hand, while ODIS attends to the correct target object, the rugby ball, demonstrating superior object-level representations.

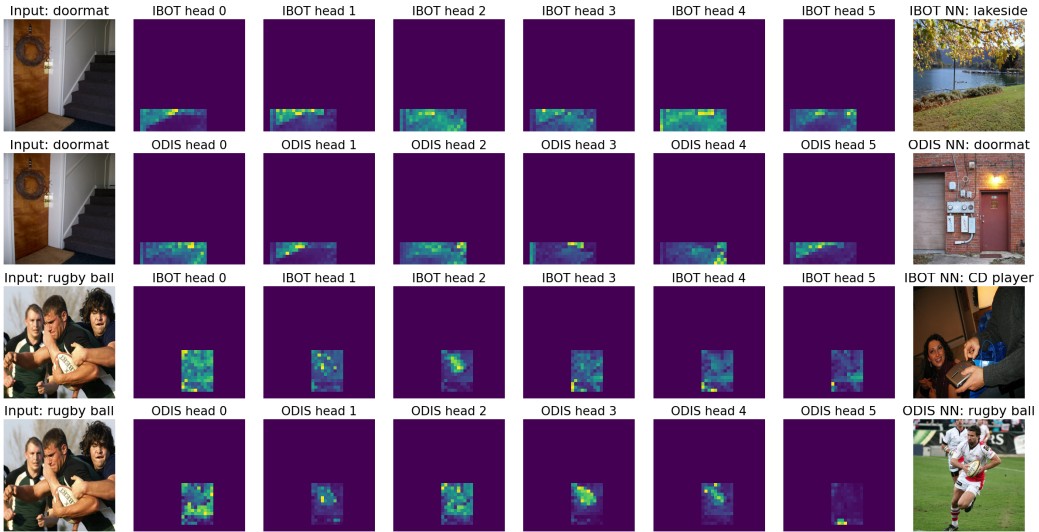

Figure 11: An extended version of Fig. 10, where all attention heads are visualized.

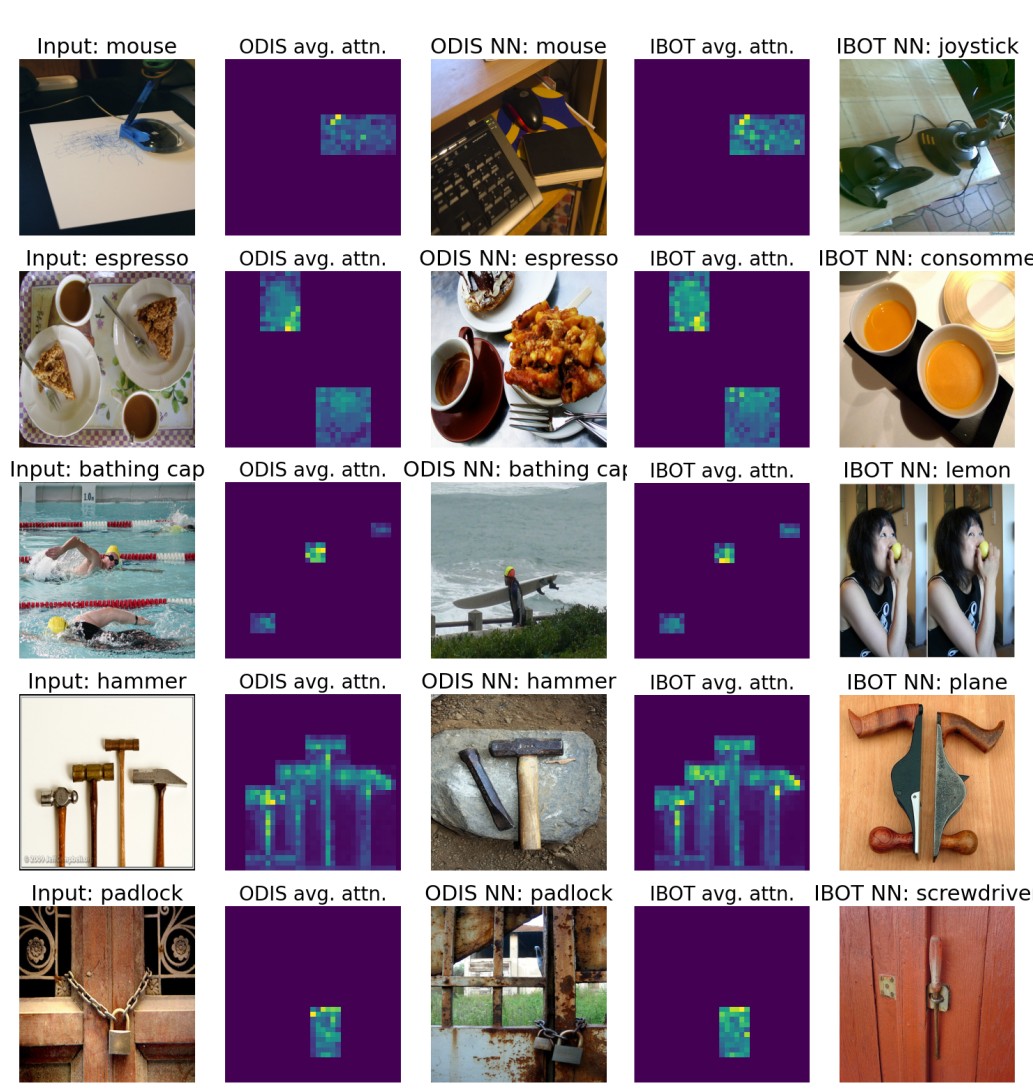

Figure 12: Additional examples showing iBOT's failure despite masked attention.

