# OpenReview forum: "Object-level self-distillation with bounding-box weak supervision improves vision pretraining"
_ICLR.cc/2026/Conference — Submitted to ICLR 2026_

### Official Review · Reviewer_B9bZ · 2025-10-31

**Soundness:** 1
**Presentation:** 2
**Contribution:** 1
**Rating:** 0
**Confidence:** 5

**Summary:**

This paper proposes using object bounding boxes to guide the learning process in self-distillation frameworks such as DINO and iBoT: Instead of using random crops, the paper uses annotated bounding boxes as image crops. In order to build bounding boxes-aware representation, the paper employs masked attention in each self-attention module to limit the attention of the [OBJ] token only to patches appearing in the bounding boxes.

The paper shows in experiments that using bounding boxes during training, the proposed method ODIS outperforms DINO and iBoT.

**Strengths:**

The paper provides evidences showing that using annotated bounding boxes to guide model training in self-distillation framework like DINO and iBoT brings some benefits.

**Weaknesses:**

In the current form, I am not sure that the paper brings any new insights into the literature. it is widely expected that using annotated bounding boxes -- which are expensive to obtain, several times more expensive than class label annotations -- would improve the model performance compared to models trained without using any annotations. The comparisons included in the paper are not fair at all.
If the central claim is "pretraining pipelines should default to using bounding boxes" (L.306) with the goal of making the best model possible regardless of the annotation cost, the paper should demonstrate the method's effectiveness at a much larger scale and compare to the best models such as SigLIP, DINOv3, etc. At this scale, the results presented in the paper look trivial.

The idea of the paper is very close to Mishra et al. (https://arxiv.org/pdf/2112.00319) which stays in the self-supervised setting by using unsupervised object discovery methods to obtain the boundign boxes. The paper could consider using a similar approach for generating bounding boxes.

I am not sure the 12 fine-grained datasets considered in section 5.2 could be considered as "out-of-distribution" with respect to ImageNet. There is a significant overlap in visual concepts between these datasets and ImageNet.

**Questions:**

I encourage the authors to consider using bounding boxes generated in an unsupervised manner to better demonstrate the method's effectiveness, or attempt at building the best possible model in a very large scale.

---

> ### Author Response · Authors · 2025-11-22
>
> We thank the reviewer for their time and for engaging with our work. We respectfully disagree with several aspects of the assessment and welcome the opportunity to clarify our scope, method, and evidence.
>
> Our contribution is straightforward: **ODIS integrates object‑level weak supervision into ViT self‑distillation by combining (i) object‑aware cropping and (ii) masked attention that restricts an [OBJ] token to attend only to patches within a bounding box**. The supervision we use is in the form of **class‑agnostic** boxes that are either **already available in common datasets** or **automatically obtainable at low marginal cost** from modern proposal generators. With this mechanism, we observe **consistent, non‑trivial** gains over strong baselines (DINO/iBOT).
>
> We emphasize that the assigned **soundness score 1/4** does not reflect the rigor of our experimental methodology. The paper follows standard SSL practice and emphasizes controlled comparisons: identical data, schedules, architectures, and evaluation protocols, differing only in the components needed for ODIS. We evaluate across **multiple architectures (ViT‑S/B/L), datasets (ImageNet‑1K, COCO)**, and **protocols** including k‑NN, linear probing, few‑shot transfer, broader transfer benchmarks, and dense prediction, and report **targeted ablations** (Table 4/5).
>
> Below, we address the reviewer’s specific concerns on (i) “widely expected” benefits, (ii) scale and comparisons to very large models, (iii) fairness and annotation cost, (iv) relation to [Mishra+22], and (v) the “out‑of‑distribution” phrasing.
>
> ---
>
> **(i) “It is widely expected that using annotated bounding boxes would improve performance”**
>
> We agree that the *intuition* is plausible; what has been missing in the self‑distillation ViT literature is a **mechanism** and **evidence** showing how to make boxes reliably improve DINO/iBOT‑style training. Our contribution is not the observation that “boxes help” in the abstract, but a concrete and simple way to **inject object cues into self‑distillation**, via **object‑aware crops** and **masked attention**, together with systematic and convincing experiments demonstrating consistent gains.
>
> Crucially, **naive uses of boxes do not reproduce our results**. In ablations (see section “On novelty vs. Mishra”), we compare (i) simply using box crops, (ii) using dilated box crops [Mishra+22], (iii) pooling patch features within boxes [Henaff+21], and (iv) ODIS. ODIS outperforms other methods with object-level weak-supervision. Hence, the **masked‑attention design is necessary**.
>
> Thus, the claim that improvements are “widely expected” does not diminish the contribution: our work **turns expectation into a reproducible method for modern self‑distillation**, specifying how to exploit object cues and verifying that this design consistently outperforms strong baselines including [Mishra+22]-style cropping.
>
> ---
>
> **(ii) On scale and trivial results**
>
> The reviewer dismisses our results as "trivial" and demands we "demonstrate the method's effectiveness at a much larger scale" by comparing it to "SigLIP, DINOv3, etc.". DINOv2 uses:
> * 142M images vs. our 1.2M (100x more data)
> * 1B parameters vs. our 307M (3x more parameters)
> * Massive compute budgets with extensive filtering pipelines
>
> **Our contribution is a training methodology, not a scaling study**. We introduce masked attention and object-aware cropping as techniques that improve over strong baselines (DINO, iBOT) at controlled, comparable scale. To ensure a fair test of the method itself, we keep data, schedules, architectures, and evaluation protocols matched, changing only the components needed to introduce ODIS. This follows standard practice for methodological research papers. Under these controlled conditions, the improvements are substantive and consistent, not “trivial”:
> * **+1.6% k-NN** (ViT-L) over iBOT, compared to iBOT's celebrated **+1.0%** improvement over DINO
> * **+4.0% dense NN retrieval** on PASCAL VOC (Table 6)
> * **+2.6%** on ADE20K segmentation (Table 6)
> * **+1.7% average** across 10 transfer datasets (Table 2: 9/10 datasets improved)
> * Gains that **scale with model size**: +0.7% (ViT-S) $\rightarrow$ +1.2% (ViT-B) $\rightarrow$ +1.6% (ViT-L)
>
> **For reference, we remind that iBOT improved about 1.0% over DINO.**

---

> > ### Author Response · Authors · 2025-11-22
> >
> > **(iii) On fairness and annotation cost**
> >
> > The reviewer claims our comparisons are "not fair at all" but we do **not** collect any additional labels. Instead, we rely on supervision that is **already available** in standard datasets or can be **generated automatically at negligible marginal cost**. Off‑the‑shelf proposal generators (e.g., YOLO, MAVL) produce boxes approximately in **1.5%** of our pretraining wall‑clock.
> >
> > All comparisons are **apples‑to‑apples**: data, schedules, architectures, and evaluation protocols are held constant; the only change is replacing random crops with object‑aware crops and enabling masked attention, so that the effect of ODIS is isolated. This experimental setup directly supports our main claim: using object-level weak-supervision improves vision pretraining.
> >
> > In addition, we provide ablations below for different ways of using object-annotations during pretraining (see section “On novelty vs. Mishra”) where we (i) simply replace random crops with box crops, (ii) use dilated box crops [Mishra+22], or (iii) pool patch features within boxes [Henaff+21], the improvements are smaller compared to our method.
> >
> > ---
> >
> > **(iv) On Novelty vs. [Mishra+22]**
> >
> > We appreciate the pointer to [Mishra+22] and clarify how our setting and contributions differ in scope and mechanism.
> >
> > **[Mishra+22]:**
> > 1. **Modeling**: Uses **ResNet** architecture and introduces **object-aware cropping**, within a SimCLR-style **contrastive learning** framework.
> > 2. **Evaluation**: Primarily on dense tasks such as object detection on COCO and semantic segmentation on PASCAL VOC with full fine-tuning.
> > 3. **Training objective**: Adds heuristic auxiliary objectives for rotation and localization prediction.
> >
> > **ODIS (ours):**
> > 1. **Modeling**: Considers **vision transformer** and introduces **masked attention**, within a **self-distillation** (teacher-student) framework.
> > 2. **Evaluation**: Primarily on frozen image embeddings on a convincing set of standard benchmarks on IN1k and 12 transfer learning datasets as in SimCLR, DINO and iBOT papers.
> > 3. **Training objective**: Keeps standard, state-of-the-art iBOT/DINOv2 objective, changes the granularity of the loss via masked-attention.
> >
> > To isolate the effect of different ways to exploit the bounding boxes, we re‑implemented common object‑level variants within the **same iBOT backbone, same data**, and **same schedule** (ImageNet‑1k, 300 epochs): (i) simple box crop + standard augmentation, (ii) dilated box crops as in [Mishra+22], and (iii) linear pooling of patch embeddings within boxes [Henaff+21]. ODIS delivers the strongest k‑NN accuracy:
> >
> > | Method | kNN |
> > | --- | --- |
> > | iBOT + crop + augment | 72.2 |
> > | iBOT + dilated crop + augment [Mishra+21] | 72.1 |
> > | iBOT + linear pool [Henaff+21] | 59.5 |
> > | ODIS | **72.7** |
> >
> > These results indicate that cropping alone (with or without dilation) does not account for our gains; the masked‑attention integration is the key difference.
> >
> > In conclusion, despite the fact that both works explore object cues, **ODIS addresses a different, and currently dominant, training regime (ViT self‑distillation)**, introduces a **new mechanism (masked attention with [OBJ])**, and demonstrates consistent improvements on standard benchmarks across model sizes.
> >
> > [Henaff+21]: Efficient Visual Pretraining with Contrastive Detection
> >
> > ---
> >
> > **(v) “Out-of-Distribution” datasets**
> >
> > We thank the reviewer for pointing this out. We agree that several of the 12 fine‑grained datasets overlap with ImageNet categories. Our intent, however, was to evaluate transfer, not strict statistical OOD. We will revise the wording accordingly and refer to this as transfer benchmarks.
> >
> > ---
> >
> > **(Q) Suggestion to use bounding boxes generated in an unsupervised manner**
> >
> > This is exactly what we show with automatically generated proposals in Table 4. Importantly, even with automatically generated (imperfect) boxes, ODIS outperforms iBOT by ~0.3-0.4% (Table 4).

---

### Official Review · Reviewer_54D9 · 2025-11-01

**Soundness:** 3
**Presentation:** 3
**Contribution:** 2
**Rating:** 6
**Confidence:** 3

**Summary:**

This paper proposes ODIS (Object-level Self-Distillation) for self-supervised learning in multi-object scenes, where DINO leads to inconsistent learning signals.
ODIS introduces two main components: (1) object-aware cropping that uses bounding boxes to ensure teacher and student views depict the same object, and (2) masked attention that constrains the [OBJ] token to attend only to patches within the object boundary. Experiments demonstrate improvements over strong baselines across ImageNet k-NN classification, linear probing, and transfer learning tasks using both ground-truth and automatically extracted bounding boxes from YOLO and MAVL.

**Strengths:**

- The paper identifies a limitation in current self-distillation methods where random crops in multi-object scenes can lead to semantic inconsistency between teacher and student views. The issue is meaningful and worth exploring.
- The object-aware cropping and masked attention mechanisms are technically sound and plausibly address the identified problem. The integration with existing methods like iBOT is clean and well-designed.
- Results show consistent improvements across multiple benchmarks and different model sizes (ViT-S, ViT-B, ViT-L).
- The paper is well written and easy to follow.

**Weaknesses:**

- While scaling is compute-intensive, scalability is increasingly expected for modern self-supervised learning approaches. The method would be significantly more compelling if its scalability were demonstrated on large, multi-object datasets such as SA-1B, OpenImages, or Objects365. Such experiments would greatly strengthen the significance and practical impact of this work, but are unfortunately absent in the current submission.
- Can the authors provide empirical comparisons to other multi-object self-supervised learning methods, such as Odin, SelfPatch, SlotMIM, and CAPI? Including these baselines would help clarify the relative advantages and limitations of the proposed approach.

References:
- [Odin] Object discovery and representation networks
- [SelfPatch] Patch-level Representation Learning for Self-supervised Vision Transformers
- [SlotMIM] A Data-Centric Revisit of Pre-Trained Vision Models for Robot Learning
- [CAPI] Cluster and Predict Latent Patches for Improved Masked Image Modeling

**Questions:**

see weaknesses

---

> ### Author Response · Authors · 2025-11-22
>
> **(W1) Scaling pretraining to large, multi-object datasets**
>
> Thanks for the insightful comment. We agree with the reviewer that scaling our method to large, multi-object datasets such as SA-1B, OpenImages or Objects365 is important. A similar point is also raised by Rev. #98h1 (see W5).
>
> In this direction, we provide two partial results:
> 1. As many web images are scene-centric, we provide an ablation study on the scene-centric dataset, COCO, (even though it is still curated and labeled). We report IN-1k kNN results in Table 1 where ODIS outperforms iBOT embeddings measured by kNN accuracy in IN1k.
> 2. Such a scaling study would require extracting class‑agnostic boxes via a foundation model. In the paper we already show that automatically generated proposals (YOLO/MAVL) suffice and still yield gains over iBOT (Table 4), directly addressing the box‑free setting where no annotations are available.
>
> Due to constraints on compute resources, we leave full exploration of this direction for future work (on the current smaller scales, our research already took ~50k gpu hours).
>
> **(W2) Additional multi-object self-supervised learning methods, such as Odin, SelfPatch, SlotMIM, and CAPI**
>
> We thank the reviewer for pointing out these multi-object baselines. Below we clearly note where direct comparisons are possible or not, and summarize the results.
>
> 1. ODIN
>
>     ODIN trains ResNet‑50 / Swin on ImageNet‑1K for 1000 epochs and evaluates primarily via full fine‑tuning on COCO detection and VOC segmentation. Code/checkpoints are not publicly released, and their evaluation differs from our frozen‑feature protocol. Consequently, a fair, direct comparison is not currently possible.
>
> 2. SelfPatch
>
>     SelfPatch trains ViT‑S/16 on ImageNet‑1K for 200 epochs using 2 global + 8 local crop. It reports ~20% longer time per epoch than DINO in their Table 1 and suggests to compare 300-epoch DINO and 200-epoch SelfPatch checkpoints. However, they also use 2 global and 8 local crops ($(2 + (96/224)^2 * 8) \approx 3.5$), resulting in an effective pre-training epoch of ~3.5 x 300 = 1050. Since our method uses 10 local crops (as DINO/iBOT), it reaches the same effective pre-training epochs in 1050 / 3.84 ~ 273. We report the closest available checkpoint (epoch 250) and evaluate frozen features on the dense NN task on PASCAL VOC and ADE20k:
>
>     | Model | VOC mIoU | ADE20k mIoU |
>     | --- | --- | --- |
>     | SelfPatch | 54.9 | 20.4 |
>     | ODIS | **61.6** | **22.9** |
>
> 3. SlotMIM.
>
>     SlotMIM trains ViT‑B/16 on ImageNet‑1K for 400 epochs and evaluates mainly with full fine‑tuning. Since a public checkpoint is available, we evaluate ImageNet‑1K k‑NN on frozen features:
>     | Model | IN1k kNN |
>     | --- | --- |
>     | SlotMIM | 68.3 |
>     | ODIS | **78.3** |
>
> 4. CAPI
>
>     CAPI trains ViT‑L/14 on ImageNet‑1K, and reports results on on IN1k, ADE20k and PASCAL VOC based on training lightweight prediction heads on frozen features, similar to our paper. Hence, we directly compare our results with their result tables:
>
>     | Model | IN1k kNN |
>     | --- | --- |
>     | CAPI - avg. pool | 68.3 |
>     | CAPI - pred. pool | 68.3 |
>     | ODIS | **79.6** |
>
>     | Model | IN1k Linear Head |
>     | --- | --- |
>     | CAPI - all patch representations + attentive probing | 82.9 |
>     | CAPI - avg. pool + linear | 77.1 |
>     | CAPI - pred. pool + linear | 81.1 |
>     | ODIS - linear | 81.6 |
>
>     | Model | VOC mIoU | ADE20k mIoU |
>     | --- | --- | --- |
>     | CAPI - kNN | 60.7 | 29.2 |
>     | ODIS - dense NN | **72.6** | **31.0** |
>     | CAPI - linear | 69.7 | 34.4 |
>     | ODIS - linear | **77.2** | **34.9** |
>
> For ODIN and other works reported only under full fine‑tuning and without public checkpoints, a direct apples‑to‑apples comparison is not feasible at present. Where frozen‑feature comparisons are possible (SelfPatch, SlotMIM via our evaluation; CAPI via their reported protocol), ODIS consistently matches or exceeds the proposed baselines.

---

### Official Review · Reviewer_98h1 · 2025-11-02

**Soundness:** 3
**Presentation:** 3
**Contribution:** 3
**Rating:** 6
**Confidence:** 4

**Summary:**

This paper proposes Object-level Self-Distillation (ODIS), a method that brings object-level supervision into self-distillation without requiring class labels. The key idea is to ensure that the student and teacher networks distill knowledge from the same semantic object, rather than arbitrary random crops that may contain different entities.
Two technical contributions define ODIS:Uses bounding boxes during training to repeatedly sample two augmentations until both the teacher and student views include the same object region;Introduces a dedicated [OBJ] token, guided by a binary attention mask derived from the bounding boxes, so that [OBJ] aggregates information only from patches inside the object region.
With this setup, ODIS explicitly shifts the distillation granularity from image-level to object-level. Experiments on ImageNet-1K and COCO show consistent gains over DINO and iBOT across multiple tasks.
At inference, the paper further explores how to utilize bounding boxes (blackout, cropping, masked attention) and finds that ODIS performs best when inference masking aligns with its training-time attention mechanism.

**Strengths:**

1.Well-motivated problem: The authors correctly identify a key limitation of existing image-level self-distillation (e.g., DINO, iBOT): random cropping may cause student and teacher to view different semantic entities, leading to inconsistent or entangled representations. ODIS directly resolves this by enforcing object-level alignment.
2.Simple yet effective method: The integration of object-aware cropping and masked attention is conceptually clean and practically compatible with existing self-distillation frameworks. It only requires bounding boxes and minimal architectural changes to ViT, making it accessible for replication.
3.Comprehensive experiments:
On ImageNet and COCO, ODIS outperforms DINO/iBOT on all image-level evaluation metrics (Table 1).
On 10 transfer benchmarks, ODIS yields an average +1.7 pp improvement in linear probing (Table 2).
On retrieval tasks (ROxf/RPar), it matches or surpasses DINO and significantly outperforms iBOT (Table 3).
On dense tasks (VOC/ADE20K/COCO-Stuff), both dense k-NN and linear segmentation improve notably (Fig. 5).
Even with noisy bounding boxes from YOLO/MAVL, ODIS still beats baselines (Table 4).
4.Insightful inference analysis: Figure 6 and Section 5.3 show that using bounding boxes during evaluation benefits all models, but ODIS benefits the most, validating its object-centric pretraining.
5.Practical implications: The paper highlights that object annotations—often freely available—can substantially enhance self-distillation without labels. This could inspire broader adoption of “weakly spatially supervised” SSL paradigms.

**Weaknesses:**

1.No training-time comparison for “cropping only” vs “masked attention.”
Figure 6 compares blacking-out, cropping, and masked attention at inference time, but not during training. An ablation with “object-aware cropping only (no masked attention)” under the same compute budget would clarify the independent contribution of each component.

2.Computation and efficiency not quantified.
The method involves up to 20 resampling trials to ensure box hits and uses masked multi-head attention in every transformer layer. The paper does not report training throughput (images/s), memory usage, or wall-clock time compared to DINO/iBOT. This makes it hard to assess efficiency trade-offs.

3.Limited analysis of bounding box quality and quantity.
Although Section 5.3 uses YOLO/MAVL for pseudo-boxes, there is no systematic study of performance vs. box quality (IoU thresholds, jitter, missing/false boxes, small objects). Such an analysis would strengthen claims of robustness.

4.Missing comparison to stronger modern baselines.
While DINO/iBOT are appropriate, recent methods such as DINOv2 or MAE + DINOv2 hybrids are omitted. Even a smaller-scale controlled comparison would clarify whether ODIS still holds advantages beyond scale improvements.

5.Scalability to unlabeled, box-free datasets not fully explored.
The method relies on bounding boxes. The paper notes that auto-generated boxes help, but large-scale web datasets often lack or contain noisy localization. Quantifying cost, bias, and privacy aspects of large-scale box generation would help position the method’s practicality.

6.Limited component-wise ablations.
Appendix E only reports ablations on the auxiliary image-level term Li and mask source. More detailed component ablations (cropping-only, masking-only, both) and sensitivity studies (teacher temperature, EMA coefficient, mask ratio) are missing.

**Questions:**

1.What are the throughput, GPU memory, and training time per epoch compared to DINO/iBOT under identical batch and resolution? How much extra overhead is caused by 20× resampling?

2.Have you tried synthetic perturbations to bounding boxes (e.g., random shifts, scaling, false positives)? How robust is ODIS under such distortions?

3.Why does training only sample one bounding box at a time? Have you tried using multiple [OBJ] tokens simultaneously for multi-object distillation?

4.Could you provide an ablation separating the effect of object-aware cropping and masked attention, trained under equal compute? This would reveal whether the main gains come from the attention mechanism or simply better aligned crops.

5.Beyond frozen features, how does ODIS perform when fine-tuned on detection or segmentation tasks (e.g., Mask R-CNN, Deeplab)? This would concretely demonstrate the value of object-level pretraining.

6.Please report or analyze the effect of teacher/student temperatures, EMA decay λ, and whether [OBJ] and [CLS] heads share parameters.

---

> ### Author Response · Authors · 2025-11-21
>
> **(W1 + Q4) No training-time comparison for “cropping only” vs “masked attention.”**
>
> We thank the reviewer for pointing this comparison out. We trained three variants for 300‑epochs on ImageNet‑1K, and report k‑NN accuracy (Edit: There was a typo for kNN value of 'Both'. We fixed it):
> | Variant | kNN |
> | --- | ----------- |
> | Cropping-only | 72.5 |
> | Masked-attention-only | 71.9 |
> | Both | 72.7 |
>
> The results suggest that cropping‑only captures most of the gain by ensuring teacher and student view the same object, yielding a strong learning signal. Masked‑attention‑only helps less because, without view alignment, the [OBJ] token can still receive inconsistent supervision across views. Combining both is best: masked attention sharpens object‑centric aggregation and reduces cross‑object leakage on top of the aligned crops, giving a further improvement (+0.2 pp) over cropping‑only.
>
> **(W2 + Q1) Computation and efficiency not quantified.**
>
> We ran ViT-S for ODIS and iBOT in IN1k for 5 epochs. On average, per epoch, iBOT took 10:40 while ODIS took 10:48: leading to a 1% increase overhead that captures all ODIS‑specific costs (view resampling + masked attention). This results in training throughputs of 2000 img/s vs. 1975 img / s during training. Memory usage is essentially unchanged (~20Gb per gpu for both iBOT and ODIS): the binary object mask is $H \times W \times 1$ and is applied to attention logits; it is negligible compared to ViT feature tensors (e.g., 384‑dim embeddings in ViT‑S).
>
> **(W3) Limited analysis of bounding box quality and quantity.**
>
> Thanks for raising this important ablation, which we had overlooked. Below we present a brief analysis of the bounding boxes and how the overall performance changes by certain properties of bounding boxes.
>
> 1. A comparison of ground truth boxes vs YOLO- and MAVL-generated boxes:
>
>     As suggested by the reviewer, we quantify the quality of bounding boxes by computing their IoU with the ground truth boxes. For YOLO, we found
>
>     | Model | mean IoU | median IoU | %img IoU>0.3 |  %img IoU>0.5 | %img IoU>0.7 | %img IoU>0.9 |
>     | --- | --- | --- | --- | --- | --- | --- |
>     | YOLO | 0.51 | 0.59 |  61% | 54% | 48% | 28% |
>     | MAVL | 0.61 | 0.65 | 81% | 63% | 47% | 23% ||
>
>     Overall, these findings indicate that MAVL boxes are of higher quality. This could be the reason why pretraining on MAVL leads to slightly better kNN accuracy over YOLO.
>
> 2. Confidence score analysis:
>
>    Along with bounding box coordinates, MAVL returns a confidence score for bounding boxes. We perform two 150-epoch training runs with bounding boxes above a confidence score of 0.25 and 0.6. We discovered that the higher threshold leads to a 0.2 percent increase in kNN accuracy. Yet, a further increase in the confidence score threshold to 0.8 caused a slight decrease (0.05 percent). We believe this could be because of a trade-off between the box count and quality: lowering the threshold causes lower-quality boxes, whereas increasing the threshold naturally leads to a smaller fraction of images with bounding box annotations.
>
> **(W4) Missing comparison to stronger baselines**
>
> We thank the reviewer for raising this point. Direct, number‑to‑number comparisons with DINOv2/MAE hybrids would largely conflate scale and recipe changes with the method under test. In particular, DINOv2 differs in ways that make a “drop‑in” comparison unfair at our scale: (i) it distills from a ~1.1B‑parameter ViT‑g into smaller models, (ii) large teacher model is trained on a curated 142M‑image corpus (LVD‑142M) rather than ImageNet‑1K; and (iii) it incorporates multiple large‑scale recipe changes (e.g., data filtering, teacher scheduling, augmentation and regularization tweaks) that are tuned for that regime.
>
> By contrast, our experiments keep data, architecture, and training schedule fixed and modify only two components: object‑aware cropping and masked attention on top of strong, widely used baselines (DINO/iBOT). This is standard practice for methods papers and allows a clean attribution of the observed improvements to ODIS.
>
> Importantly, ODIS is orthogonal and modular: the same mechanism (aligned crops + [OBJ] attention masking) can be integrated into larger‑scale pipelines, including DINOv2 or MAE. Our claim is therefore not that ODIS surpasses billion‑parameter, hundred‑million‑image systems, but that given any self‑distillation pipeline at a fixed budget, injecting object‑level alignment yields consistent gains.

---

> > ### Author Response · Authors · 2025-11-21
> >
> > **(W5) Scalability to unlabeled, box-free datasets not fully explored**
> >
> > Thanks for the insightful comment. We agree with the reviewer that scaling our method to unlabeled web-scale large datasets is important. A similar point is also raised by Rev. #54D9 (see W1).
> >
> > In this direction, we provide two partial results:
> > 1. As many web images are scene-centric, we provide an ablation study on the scene-centric dataset, COCO, (even though it is still curated and labeled). We report IN-1k kNN results in Table 1 where ODIS outperforms iBOT embeddings measured by kNN accuracy in IN1k.
> > 2. Such a scaling study would require extracting class‑agnostic boxes via a foundation model. In the paper we already show that automatically generated proposals (YOLO/MAVL) suffice and still yield gains over iBOT (Table 4), directly addressing the box‑free setting where no annotations are available.
> >
> > Due to constraints on compute resources, we leave full exploration of this direction for future work (on the current smaller scales, our research already took ~50k gpu hours).
> >
> > **(W6) Limited component-wise ablations**
> >
> >
> > We expanded the ablations to disentangle component effects and hyperparameter sensitivities.
> >
> > 1. Factorized components.
> > The requested cropping‑only vs. masking‑only vs. both comparison is reported in section #W1+Q4. In brief, cropping provides the alignment signal, masked attention sharpens object aggregation, and combining both yields the best performance.
> >
> > 2. Mask ratio sweep (IN1k, 150 epochs, IN1k):
> >
> >     We provide kNN results for mask ratio sweep in the table below:
> >     | Mask ratio | kNN |
> >     | --- | ------ |
> >     | 0.1 | 71.1 |
> >     | 0.5 | 71.8 |
> >     | 0.8 | 71.7 |
> >     | 1.0 | 71.9 |
> >
> >     A very low mask ratio (0.1) is insufficient for learning stable object localization. Using masks most of the time (0.5–1.0) performs on par. Additionally, we ablated a mask ratio scheduler for the masked attention component with a linear warm-up from 1.0 to 0.3 in epochs 50-150.
> >
> > 3. Mask‑ratio scheduling.
> >
> >     We also tested a linear schedule for using masked attention component from $1.0 \rightarrow 0.3$ over epochs 50-150. This improved k‑NN from 71.9 → 72.2, suggesting that moderating the weak supervision ratio over training can yield small but consistent gains.
> >
> > 4. Alternatives for using object cues.
> >
> >     As detailed in section “On Novelty vs. [Mishra+22] in response to reviewer #B9bZ”, we compare ODIS with three alternatives integrated into the same iBOT backbone on 300‑epoch ImageNet‑1K pretraining: (i) simple box crop + standard augmentation, (ii) dilated box crops [Mishra+22], and (iii) linear pooling of patch embeddings within boxes [Henaff+21]. ODIS attains the strongest k‑NN accuracy.
> >
> >
> > 5. Other sensitivities
> >
> >     At the early phase of model development, we conducted small sweeps around the teacher temperature and EMA coefficient used in DINO/iBOT and did not observe meaningful improvements; we therefore retain the standard settings from those baselines.
> >
> > **(Q6) Teacher/student temperatures, EMA decay $\lambda$, and whether [OBJ] and [CLS] heads share parameters**
> >
> > We follow iBOT: (i) teacher temperature is linearly warmed-up from 0.04 to 0.07 in 30 epochs, (ii) student temperature is 0.1, (iii) EMA decay $\lambda$ follows a cosine schedule from 0.996 to 1. The heads share parameters.
> >
> > **(Q2) Synthetic perturbations to bounding boxes (e.g., random shifts, scaling, false positives)**
> >
> > Thank you for this robustness question. We have not yet evaluated random shifts, scalings, or explicit false‑positive boxes. We agree that these tests would add valuable analysis. However, they are orthogonal to our core claim, and we are primarily limited by compute to include them in this submission.
> >
> > **(Q3) Multiple [OBJ] tokens simultaneously for multi-object distillation**
> >
> > We explored a multi‑object variant with multiple [OBJ] tokens during early development (preliminary 40‑epoch COCO runs). For example, we measured performance on PASCAL VOC dense NN segmentation task across models with 1, 2 and 4 [OBJ] tokens during pretraining. Our experiments showed that multiple-token variants underperformed the single‑object version: (i) 1-[OBJ]-mIoU: 27.1, (ii) 2-[OBJ]-mIoU: 25.0, (iii) 4-[OBJ]-mIoU: 22.8; and added non‑trivial complexity (extra tokens/masks per layer and loss terms). A likely reason might be non-contrastive loss pushing all [OBJ] tokens to the same embedding (unless the loss is redesigned).
> >
> > Given these observations, we chose the simpler and better model for larger scale training. We agree that multi‑object distillation is promising; it appears to require additional design beyond a naive multi‑token drop‑in, and we view it as an interesting avenue for future work.
> >
> > **(Q5) ODIS fine-tuned on detection or segmentation tasks**
> >
> > We are working on these tasks, we hope to finish them by December 3 and report results.

---

> > > ### Author Response · Authors · 2025-12-04
> > >
> > > Regarding full finetuning, we ran full fine-tuning scripts for semantic segmentation with ViT-S backbone on ADE20k and Pascal VOC datasetets for our method and iBOT. We experimented on 4 different learning rates (8e-6, 1e-5, 3e-5, and 8e-5), and used the UPerNet based segmentation head employed by the iBOT paper. We report the validation score for the best learning rate and epoch:
> > > * ADE20K:
> > >   * iBOT 46.3
> > >   * ODIS 46.7
> > > * PascalVOC:
> > >   * iBOT 82.4
> > >   * ODIS 82.5
> > >
> > > Results suggest that ODIS backbone performs slightly better than iBOT on these tasks. These results are in line with general trend validated in many papers (Hummingbird [Balazevic+23], CRIBO [Lebailly+23], iBOT [Zhou+21], DINOv2 [Oquab+23]): The performance in frozen features are in general a good proxy for full finetuning performance, that is, the performance ranking mostly stays the same between evals (i) on frozen features, and (ii) with full fine-tuning (with the notable exception of MAE [He+21]); while evaluation on frozen features are much more compute-efficient.
> > >
> > > We continue working on full finetuning for detection tasks.

---

### Author Response · Authors · 2025-12-04

# General Response

We thank all reviewers for their careful and constructive feedback. We are encouraged by the shared recognition of **(i)** our well-motivated problem (Revs. #98h1, #54D9), **(ii)** the simplicity and effectiveness of our approach (Revs. #98h1, #54D9), **(iii)** our comprehensive experiments with consistent results (Revs. #98h1, #54D9), **(iv)** the practical implications of our work for further impact (Rev. #54D9), **(v)** the insightfulness of our inference‑time analysis (Rev. #98h1), and **(vi)** our well-written, easy-to-follow text (Rev. #98h1). Below, we provide a point‑by‑point response to all comments.

---

### Meta-Review · Area_Chair_GY7i · 2026-01-11

**Summary:**

Reviewers #98h1 and #54D9 view ODIS as a simple, well-motivated object-centric modification to ViT self-distillation with consistent gains across ImageNet/COCO transfer and robustness to noisy boxes. The main decision-driving issues are (i) whether the contribution is sufficiently novel beyond prior box-guided/object-centric SSL work, and (ii) whether the method’s practicality/impact is demonstrated beyond ImageNet-scale training (e.g., larger datasets / stronger modern baselines). Despite two supportive reviews, a high-confidence strong reject remains, and even with optimistic score shifts the paper stays below an accept bar due to limited scale and modern-baseline evidence.

**Reviewer Concerns:**

**Reviewer #98h1 (score 6, conf 4)**
Requested training-time ablation (cropping-only vs masked-attention-only vs both) was provided; efficiency/overhead was quantified (≈1% slower than iBOT in a short run; similar memory); added box-quality/quantity analysis (IoU stats; confidence threshold trade-off); added some sensitivity/implementation details; added segmentation fine-tuning results (small gains vs iBOT). However, there is no direct comparison to very strong modern pretraining baselines (e.g., DINOv2-scale recipes); robustness to synthetic perturbations (shift/scale/false boxes) not tested; large-scale/box-free web setting only partially addressed (auto boxes shown, but no web-scale study).

**Reviewer #54D9 (score 6, conf 3)**
Provided comparisons to several multi-object SSL baselines where feasible, arguing non-comparability for some (ODIN); clarified scaling limitations and gave partial evidence via COCO + auto-box setting. But, no convincing scalability demonstration on large multi-object datasets (SA-1B / OpenImages / Objects365), which the reviewer explicitly wanted; overall impact still tied to limited-scale evidence.

**Reviewer #B9bZ (score 0, conf 5)**
Authors clarified scope (method paper, controlled comparisons), responded on the annotation-cost argument by emphasizing pseudo-boxes; provided explicit novelty positioning vs Mishra et al. (and a controlled re-implementation comparison showing cropping/dilated cropping alone doesn’t match ODIS); agreed to fix “OOD” wording. Nonetheless, the reviewer’s core stance (novelty/importance insufficient unless shown at a much larger scale and/or against top-end modern models) is only partially mitigated; they may still view the contribution as incremental and the supervision assumption as too strong.

**Reviewer Scores:**

- 98h1: 6 ( Likely improvement, most concrete requests were answered with new ablations + efficiency + box analyses; remaining issues are “nice-to-have/scale”).
#54D9: 6 (added baseline comparisons help, but the key “scale to large datasets” ask remains unmet).
#B9bZ: 0  (even after rebuttal, they likely remain strongly negative given confidence=5 and their emphasis on scale/novelty).
Missing review(s): None indicated.

---

### Decision · Program_Chairs · 2026-01-26

Reject